# Weakening Neurons:
# A Newly Discovered Read-Write Functionality in Transformers with Outsize Influence

## Abstract

We introduce a new mechanistic interpretability method for gated neurons, based on the cosine similarities between their weight vectors, and use it to gain a number of novel insights into the inner workings of transformer models. First, our method allows us to discover a class of neurons – *weakening* neurons – with surprising behavior: even though there are few, they activate extremely often and have a large influence on model behavior. Second, we show that nine different LLMs have similar patterns with respect to weakening neurons: weakening neurons appear mostly in late layers whereas their counterparts, *(conditional) strengthening* neurons, are very frequent in early-middle layers. Third, weakening neurons have a strong effect on model output when gate values are negative – which is surprising since negative gate values are not expected to encode functionality. Thus, for the first time, we observe a mechanism important for transformer functionality that involves negative gate values. [1]

## 1 Introduction

Mechanistic interpretability research attempts to reverse-engineer the *mechanisms* inside neural networks, such as transformer-based (Vaswani et al., 2017) large language models (LLMs). Some of this work has addressed the interpretation of MLP sublayers, and we follow this line of research. Unlike previous work, we focus on gated activation functions (Shazeer, 2020), which are used in recent LLMs like OLMo, Llama and Gemma, but so far lack an extensive analysis from the interpretability perspective.

Much previous work analyzes neurons[2] based only on the contexts in which they activate (Voita et al., 2024) or based only on their output weights[3] (Gurnee et al., 2024). However, neither of these fully captures the *mechanisms* that neurons implement: we also have to understand the relationship between input and output behavior of neurons, what we call their read-write (RW) functionality. We put this in the center of our analysis, and propose a simple method to investigate RW functionalities: computing cosine similarities between input (reading) and output (writing) weights.

This new approach allows us to gain a number of striking novel insights into the inner workings of LLMs. In particular, we discover a small class of neurons – *weakening* neurons – with outsize influence and often surprising behavior.

Our contributions are as follows: (i) We are the first to investigate read-write behavior of gated neurons, using cosine similarities of weight vectors. (ii) Applying this method to nine LLMs, we observe universal patterns: Early-middle layers contain many *conditional strengthening* neurons, and late layers tend more towards *weakening*. (iii) Thanks to the RW perspective, we discover a small class of neurons (*weakening* neurons), that is highly influential in often surprising ways: they activate often (in the sense of having a gate value above zero), and they influence various metrics, even in earlier layers where they are very rare. (iv) We introduce a new method of conditional ablation that enables us to find *which activations* of a given neuron are responsible for a certain behavior. (v)

---

[1] We publish code at `https://anonymous.4open.science/r/RW_functionalities-4D32`.
[2] We use "neuron" to refer to a hidden dimension inside the MLP layer.
[3] We use "weight" to refer to a weight vector, not a scalar.

Applying this method to weakening neurons, we find that some of their effect is due to cases in which their gate value is negative. Thus, for the first time, we observe a mechanism involving negative values of the Swish activation function.

## 2 RELATED WORK

There is a large body of work on interpretability of transformer-based LLMs. Elhage et al. (2021) introduce the notion of residual stream. nostalgebraist (2020), Belrose et al. (2023) propose to interpret residual stream states as intermediate guesses about the next token; Rushing & Nanda (2024) discuss this as the *iterative inference hypothesis*. On a similar note, many works hypothesize that directions in model space can correspond to concepts; Park et al. (2024) discuss this as the *linear representation hypothesis*. Lad et al. (2024) define *stages of inference*. Similar to our work, Elhelo & Geva (2024) investigate input-output functionality of heads (instead of neurons).

Much research has attempted to understand individual neurons. Geva et al. (2021) present them as a key-value memory. Other neuron analysis work includes (Miller & Neo, 2023; Niu et al., 2024).

The focus on individual neurons has been criticized. Morcos et al. (2018) find that in good models, neurons are not monosemantic (but for image models, not LLMs). Millidge & Black (2022) find interpretable directions that do not correspond to individual neurons. Elhage et al. (2022) argue that interpretable features are non-orthogonal directions in model space and can be superposed. This corresponds to sparse linear combinations of neurons in MLP space. This has inspired a series of work on sparse autoencoders (SAEs), starting with Sharkey et al. (2022).

The focus on SAEs has been criticized: recent studies indicate that they do not always outperform baselines (Kantamneni et al., 2025; Leask et al., 2025; Mueller et al., 2025; Wu et al., 2025). A middle ground is possible: Gurnee et al. (2023) argue that interpretable features correspond to sparse combinations of neurons; this includes 1-sparse combinations, i.e., individual neurons. Accordingly, there is recent work on new classes of interpretable neurons, e.g. Ali et al. (2025); Zhao et al. (2025).

Several works classify neurons based on the **contexts** in which they activate (Voita et al., 2024; Gurnee et al., 2024). For example, Voita et al. (2024) find *token detectors* that suppress repetitions. Gurnee et al. (2024) also define *functional roles* of neurons based on their **output** weight vector, such as *suppression neurons* that suppress a specific set of tokens. Stolfo et al. (2024) also investigate some output-based neuron classes.

There has been less focus on the input-output perspective. Gurnee et al. (2024) compute cosine similarities between input and output weights for GPT-2 (Radford et al., 2019), but do not interpret their results. Elhage et al. (2021) mention the idea of input-output analysis (footnote 7), but do not follow up. Note that input-output analysis for gated activation functions adds complexity because, in addition to input and output weight vectors, the gating mechanism is crucial for RW functionality.

## 3 PRELIMINARIES

### 3.1 GATED ACTIVATION FUNCTIONS

We work on *gated activation functions* like SwiGLU or GEGLU (Shazeer, 2020). Gated activation functions are used widely, e.g., OLMo (Groeneveld et al., 2024) and Llama (Touvron et al., 2023) use SwiGLU, and Gemma (Gemma, 2024) uses GEGLU. Here we briefly describe SwiGLU. GEGLU replaces Swish with GELU, but is otherwise identical.

Traditional activation functions like ReLU require one weight matrix on the input side and one on the output side: The MLP outputs

$$W_{\text{out}}\text{ReLU}(W_{\text{in}}x_{\text{norm}}),$$

where ReLU is applied element-wise to each neuron (it takes a single scalar as argument).[4]

---

[4]We write $x_{\text{norm}}$ for the residual stream state before the MLP layer of interest (with layer normalization already applied, for the many models that use pre-norm).

Other traditional activation functions are $\text{Swish}(x) := x/(1 + \exp(-x))$ (Ramachandran et al., 2017) and $\text{GELU}(x) := x\Phi(x)$[5] (Hendrycks & Gimpel, 2016). Both of these can be seen as smooth approximations of ReLU. They are known to work better than ReLU, which is widely believed to be because of their good differentiability (e.g. Lee, 2023), i.e. better training dynamics.

In contrast to these traditional functions, a *gated activation function* like SwiGLU requires two weight matrices on the input side: The MLP outputs

$$W_{\text{out}} \left( \text{Swish}(W_{\text{gate}} x_{\text{norm}}) \odot (W_{\text{in}} x_{\text{norm}}) \right), \tag{1}$$

where $\odot$ denotes element-wise multiplication (a.k.a. Hadamard product).[6]

We find it more intuitive to separately consider each neuron: The neuron adds the vector

$$\text{Swish}(\langle w_{\text{gate}}, x_{\text{norm}} \rangle) \cdot \langle w_{\text{in}}, x_{\text{norm}} \rangle \cdot w_{\text{out}} \tag{2}$$

to the residual stream. Here $w_{\text{gate}}, w_{\text{in}}$ are each one of the $d_{\text{MLP}}$ *rows* of $W_{\text{gate}}, W_{\text{out}}$, and $w_{\text{out}}$ is one of the $d_{\text{MLP}}$ *columns* of $W_{\text{out}}$.[7] These weight vectors, as well as $x_{\text{norm}}$, all have dimensionality $d_{\text{model}}$.[8]

In this framework, SwiGLU can be described as a function of two scalars:

$$\text{SwiGLU}(x_{\text{gate}}, x_{\text{in}}) := \text{Swish}(x_{\text{gate}}) \cdot x_{\text{in}},$$

Unlike ReLU, gated activation functions can output arbitrary positive or negative values. For example, if $x_{\text{gate}} > 0$ and $x_{\text{in}} \ll 0$, then $\text{SwiGLU}(x_{\text{gate}}, x_{\text{in}}) \ll 0$.

## 3.2 Weight preprocessing

Our code uses TransformerLens (Nanda & Bloom, 2022). When a model is loaded, certain preprocessing steps are applied to the weights, in order to make the weights more interpretable without changing model behavior (see their documentation for details).

We propose an additional preprocessing step specific to gated activation functions: For each neuron, we multiply $w_{\text{in}}$ and $w_{\text{out}}$ by the sign of $\cos(w_{\text{gate}}, w_{\text{in}})$. See section C for our argument on why we do this and why it does not change model behavior.

## 4 Method

### 4.1 Approach

We follow the *residual stream* perspective of Elhage et al. (2021): Individual model units *read* from the residual stream and then update it by *writing* (adding to it). In the case of an MLP neuron, the scalar products $\langle w_{\text{gate}}, x_{\text{norm}} \rangle$ and $\langle w_{\text{in}}, x_{\text{norm}} \rangle$ can be thought of as *reading* how much the residual stream $x_{\text{norm}}$ conforms to the *directions* $w_{\text{gate}}$ and $w_{\text{in}}$. The neuron then *writes* a multiple of the direction $w_{\text{out}}$ to the residual stream.

A semantic interpretation is that a neuron detects a *concept* in the residual stream, and in turn also writes a concept. This semantic interpretation is not a necessary assumption for our neuron classification, but is helpful for building intuition and interpreting results.

This framework leads to our main research question: **What is the relationship between what a neuron reads and what it writes?** We take an approach based on **weights** (as opposed to activations) of **neurons** (as opposed to SAE-like features).

**Weight-based.** There are many ways to address the question; we choose a purely weight-based approach: computing the **cosine similarity of input and output weights**. This lets us understand the mathematical function that a neuron implements in terms of updates to the residual stream.

---

[5]$\Phi$ is the cumulative distribution function (cdf) of a standard normal distribution.

[6]In other works, $W_{\text{in}}$ and $W_{\text{out}}$ are also called $W_{\text{up}}$ and $W_{\text{down}}$, respectively.

[7]This is assuming the right-to-left notation of our equation (1). With left-to-right notation rows and columns are switched.

[8]Following TransformerLens (Nanda & Bloom, 2022), we write $d_{\text{model}}$ for the model dimensionality (i.e. the dimensionality of the residual stream), and $d_{\text{MLP}}$ for the hidden dimensionality of a given MLP layer.

Table 1: Six prototypical read-write (RW) functionalities. See section 4.2 for details.

| $\cos(\boldsymbol{w}_{\text{in}}, \boldsymbol{w}_{\text{out}})$ | $\lvert\cos(\boldsymbol{w}_{\text{gate}}, \boldsymbol{w}_{\text{out}})\rvert$ | $\approx 1$ (or $> 0.5$) | $\approx 0$ (or $< 0.5$) |
|---|---|---|---|
| $\approx +1$ (or $> +0.5$) | | strengthening | conditional strengthening |
| $\approx -1$ (or $< -0.5$) | | weakening | conditional weakening |
| $\approx 0$ (or $\in [-0.5, +0.5]$) | | proportional change | orthogonal output |

**Neuron-based.** This cosine similarity method could in principle also be applied to SAE-style features instead of neurons, as long as each feature has well-defined input and output weights. However, for this paper we decided to just investigate neurons, and defer a possible investigation of SAEs to future work. We do so for the following reasons: (i) As argued in section 2, individual neurons can still be a promising research direction today. (ii) For any given LLM, neurons are readily available and clearly defined, whereas researchers may have published several SAEs with different sizes and architectures – or, for less popular models, no SAE at all. (iii) We expect that findings from neurons will, to some extent, carry over to linear combinations of neurons. See section D for more on this.

In section 5 we will see that, despite being "only" weight-based and neuron-based, our method already yields striking results.

## 4.2 TAXONOMY OF RW FUNCTIONALITIES

We now think through what different combinations of weight cosine similarities would mean for neuron RW functionality, and introduce our terminology. For the moment we focus on the prototypical cases, in which cosine similarities are approximately $\pm 1$ or $0$. We present a taxonomy of these prototypical RW functionalities in table 1.

Generally, when the output weight is similar enough to (one of) the detected directions, we speak of *input manipulation*, as opposed to **orthogonal output** neurons which write to directions not detected in the input. Intuitively, input manipulator neurons *manipulate* the concept that they detect.

As special cases of input manipulation, we define: (i) **Strengthening** and **weakening** neurons: all three weight vectors are roughly collinear, and specifically $\cos(\boldsymbol{w}_{\text{in}}, \boldsymbol{w}_{\text{out}}) \approx \pm 1$. The neuron detects a direction and then adds it to / removes it from the residual stream. (ii) **Conditional strengthening / weakening** neurons: $\boldsymbol{w}_{\text{in}}$ and $\boldsymbol{w}_{\text{out}}$ are roughly collinear and $\boldsymbol{w}_{\text{gate}}$ is orthogonal to them. The neuron also strengthens / weakens the direction detected by its $\boldsymbol{w}_{\text{in}}$ vector, but will only activate *conditional on $\boldsymbol{w}_{\text{gate}}$ being present in the residual stream*. (iii) **Proportional change** neurons: $\boldsymbol{w}_{\text{out}}$ is collinear to $\boldsymbol{w}_{gate}$, but is orthogonal to $\boldsymbol{w}_{\text{in}}$. If $\boldsymbol{w}_{\text{gate}}$ is present in the residual stream, then the neuron writes a *positive or negative* multiple of this direction to the residual stream. This multiple is proportional to the presence of $\boldsymbol{w}_{\text{in}}$ in the residual stream.

These prototypical classes are limited in scope: Many cosines will not be close to $0$ or $\pm 1$. For this general case, this paper explores three options to understand neuron RW functionalities at different levels of granularity: (1) Classify neurons according to the closest prototypical case (we choose a threshold $\tau = \pm 0.5$). (2) Plot the marginal distributions of the three cosine similarities. (3) Place neurons in a scatter plot, based on their three weight cosines.

In option 1 (threshold-based classification), $\cos(\boldsymbol{w}_{\text{in}}, \boldsymbol{w}_{\text{gate}})$ may not always "match" the other two cosine similarities. Consider for example the case of strengthening: In the prototypical case with exact equalities ($\cos(\boldsymbol{w}_{\text{in}}, \boldsymbol{w}_{\text{out}}) = \cos(\boldsymbol{w}_{\text{gate}}, \boldsymbol{w}_{\text{out}}) = 1$), all three weight vectors are collinear, so we also have $\cos(\boldsymbol{w}_{\text{in}}, \boldsymbol{w}_{\text{gate}}) = 1$. But without exact equalities (if we just know $\cos(\boldsymbol{w}_{\text{in}}, \boldsymbol{w}_{\text{out}})$ and $\cos(\boldsymbol{w}_{\text{gate}}, \boldsymbol{w}_{\text{out}})$ are both above $0.5$), it does not follow that $\cos(\boldsymbol{w}_{\text{in}}, \boldsymbol{w}_{\text{gate}})$ is also above $0.5$.[9] When such a "mismatch" occurs, we prepend *atypical* to the category's name: In this example, we will

---

[9]For example, the two reading weights may be orthogonal ($\cos(\boldsymbol{w}_{\text{in}}, \boldsymbol{w}_{\text{gate}}) = 0 < 0.5$), but $\boldsymbol{w}_{\text{out}} = \boldsymbol{w}_{\text{gate}} + \boldsymbol{w}_{\text{in}}$; then $\cos(\boldsymbol{w}_{\text{in}}, \boldsymbol{w}_{\text{out}}) = \cos(\boldsymbol{w}_{\text{gate}}, \boldsymbol{w}_{\text{out}}) \approx 0.71 > 0.5$.

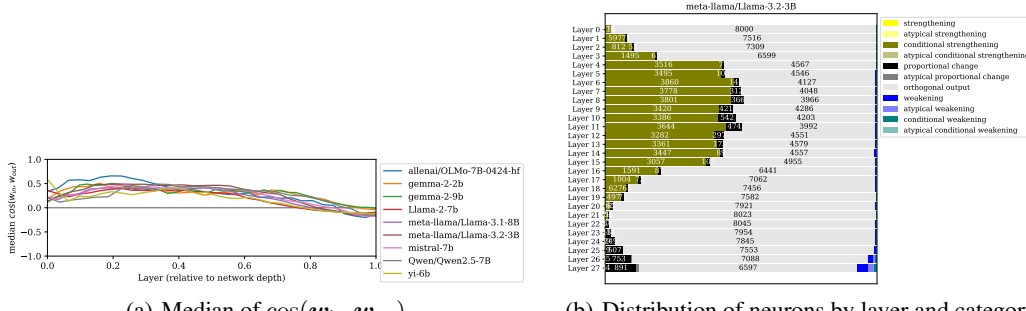

| (a) Median of $\cos(\boldsymbol{w}_{\text{in}}, \boldsymbol{w}_{\text{out}})$ | (b) Distribution of neurons by layer and category. |

Figure 1: (a) Median of $\cos(\boldsymbol{w}_{\text{in}}, \boldsymbol{w}_{\text{out}})$ by layer (x-axis) for 9 models of 2B to 9B parameters. For all models, the value is positive in the beginning and negative in the end, indicating that early-middle layers "strengthen" directions they find in the residual stream whereas later layers tend more towards "weakening" them. (b) Distribution of neurons by layer and category.

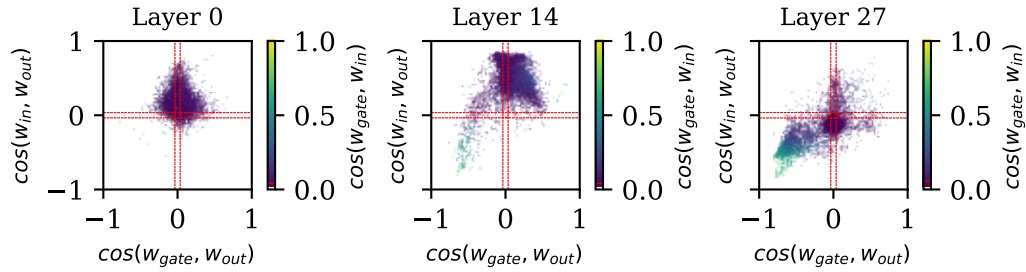

Figure 2: Fine-grained analysis of neuron RW behavior in three layers of Llama-3.2-3B, based on the configuration of their three weight vectors in parameter space. Each subplot represents a layer, each dot a neuron. The red lines mark the 95% randomness regions for each of the three cosine values. (There is a dotted line for variant (i) and a dashed line for variant (ii) in section 4.3, but they are almost the same.)
We see that many neurons are outside the randomness regions, indicating that they manipulate their input in some way. Purple dots at the top of the plots are conditional strengthening neurons. Lighter dots in the bottom left corner are weakening neurons.

speak of an atypical strengthening neuron. In figure 1(b) we will see that such neurons exist, but are quite rare overall.

## 4.3 RANDOM BASELINES

Given a cosine similarity between weight vectors, we test if it is significantly different from random. To do so, we consider two random baselines: (i) i.i.d. Gaussian vector entries (as in a randomly initialized model); (ii) a layer-specific baseline based on "mismatched cosines". See section E for details. In practice, we find that both baselines give quite similar 95% randomness ranges.

## 5 WHERE TO FIND WEAKENING NEURONS

In this section we compute cosine similarities of neuron weights as described in section 4, to investigate which RW functionalities actually appear in LLMs, and in which layers. Strikingly, our results are **consistent across models**. Across models, we find that there is a small but (as we will see later) influential number of **weakening neurons**, mostly in late layers. Other RW functionalities appear in other ranges: in particular, early-middle layers of all models contain a lot of conditional strengthening neurons.

Concretely, we apply our method to 12 LLMs: Gemma-2-2B, Gemma-2-9B (Gemma, 2024), Llama-2-7B, -3.1-8B, -3.2-1B, -3.2-3B (Touvron et al., 2023), OLMo-1B, OLMo-7B-0424 (Groeneveld et al., 2024), Mistral-7B (Jiang et al., 2023), Qwen2.5-0.5B, Qwen2.5-7B (Yang et al., 2024), Yi-6B (01.AI et al., 2025). These models use SwiGLU, except for Gemma, which uses GeGLU.

To demonstrate our finding in more detail, we present three representative plots. (See section J for more.) Figure 1(a) shows the median value of $\cos(\boldsymbol{w}_{\text{in}}, \boldsymbol{w}_{\text{out}})$ across all layers of the nine larger models. The common pattern is clearly visible: In early-middle layers of all models, a majority of neurons has a $\cos(\boldsymbol{w}_{\text{in}}, \boldsymbol{w}_{\text{out}})$ high above zero, indicating strengthening; in late layers, this median cosine similarity goes slightly below zero, indicating a relative majority of weakening neurons.

The other two plots focus on **Llama-3.2-3B**, but the patterns we describe are general: see section J for other models. Figure 1(b) shows RW class distribution across layers. In figure 2, we plot the distribution of neurons in a few selected layers, by displaying each neuron as a point with $\cos(\boldsymbol{w}_{\text{gate}}, \boldsymbol{w}_{\text{out}})$ indicated on the x-axis, $\cos(\boldsymbol{w}_{\text{in}}, \boldsymbol{w}_{\text{out}})$ on the y-axis and $\cos(\boldsymbol{w}_{\text{gate}}, \boldsymbol{w}_{\text{in}})$ as its color.

**Input manipulation.** First, we see that a large proportion of neurons are input manipulators (i.e., they are not orthogonal output neurons): In figure 1(b), these are 25% of all neurons, and as much as 50% in early-middle layers (layers 7–11[10]). What is more, figure 2 shows that even the neurons classified as "orthogonal output" often belong to clusters that are centered above/below the horizontal line. Their weight cosine similarities often exceed the significance threshold. E.g., in layer 14, there are many neurons whose $\cos(\boldsymbol{w}_{\text{in}}, \boldsymbol{w}_{\text{out}})$ (y-axis) is below 0.5 but above the significance threshold. This suggests that even the "orthogonal output" neurons perform input manipulation to some extent.

**Different RW functionalities.** Weakening neurons represent a large share of the (relatively few) input manipulators in late layers. They form a somewhat separate cluster in figure 2 (in the bottom-left corner of the rightmost subplot). Another important input manipulator class in late layers is proportional change. In contrast, across all models, early middle layers are dominated by conditional strengthening. In fact, the majority of input manipulators (more than 80% in Llama) belong to just this one class.

This general pattern of strengthening-then-weakening holds across models, as figure 1(a) shows at one glance. In figure 2 (and figure 52 in the appendix), the pattern manifests as a large cluster of neurons, centered clearly above the x-axis in most layers, but moving below it in the last few layers.

In summary, we find across models that conditional strengthening dominates in early-middle layers, but in late layers we find more weakening neurons.

## 6 Ablation experiments

Since model training produced so many input manipulator neurons, we hypothesize that they must contribute to model performance in an important way. We now test this hypothesis by ablating neurons based on their RW functionality. We find that weakening neurons have the highest effect on the metrics that we tested – this is completely unexpected since weakening neurons are a very small class of a few hundred neurons.

In the rest of the paper, we run a model on a dataset – whereas in section 5 we just applied our weight-based method from section 4. Therefore, to save resources, we focus on a single model: We choose **OLMo-7B**, because its training dataset, Dolma (Soldaini et al., 2024), is publicly available and its RW functionalities mostly follow the typical patterns. As a dataset, we use a random subset of 20M tokens from Dolma,[11] except for attribute rate, where we follow the setup of Geva et al. (2023).

### 6.1 Exploring RW classes and metrics

We run the model on our dataset and record various metrics, such as the loss. In each run we ablate a number of neurons from a different RW class, or (as a baseline) the same number of *random* neurons from the same layers. This enables us to observe the effect of various RW classes on these metrics. The baseline verifies if effects are due to the layers rather than RW classes.

---

[10]We use zero-based (Python-style) indexing throughout the paper.

[11]The size of 20M tokens follows Voita et al., 2024.

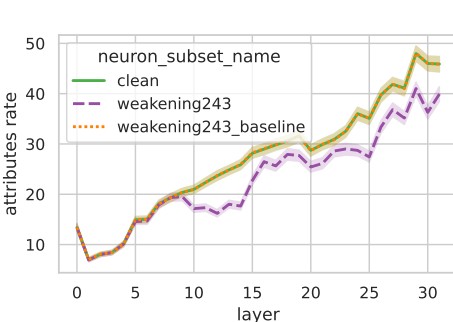

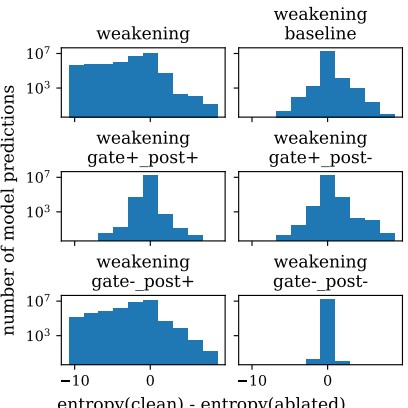

(a) Effect on attribute rate of ablating all 243 weakening neurons (weakening243), or 243 random neurons from the same layers (weakening243_baseline). These baseline neurons do not have any sizable influence. In contrast, the effect of weakening neurons is clearly visible, already from layer $\approx 10$ onward, even though weakening neurons are few and mostly in late layers. In the appendix (figures 14 to 16) we show results for other neuron classes, all of which are indistinguishable from the "clean" line.

(b) Effect on entropy of various neuron activations. E.g., in $\approx 10^5$ next-token predictions, weakening neurons decrease the entropy by about 10 nats, whereas they increase it much more rarely. "Weakening baseline" denotes random neurons from the same layers as weakening neurons. The bottom four plots describe the results of conditional ablations (section 6.2). E.g., "gate+_post+" describes the effect of those activations in which $x_{\text{gate}} > 0$ and $x_{\text{post}} > 0$.

Figure 3: Effect of zero-ablating weakening neurons.

The main metrics we consider are attribute rate (Geva et al., 2023) and entropy of the output distribution. We justify these choices in section F. We try two types of ablation: zero ablation (setting activations to zero), and mean ablation (setting them to the mean activation of the given neuron).

We find that **ablating weakening neurons has a large effect** on both metrics, and this effect is not seen with other classes or with other neurons from the same layers. The effect is clearest with zero ablation, but also present with mean ablation (see section F.4 for mean ablation results). For attribute rate, the effect is most visible in layers $\approx 10$ and onward. See figure 3(a). This is particularly interesting since there are very few weakening neurons in these early-middle layers. The case of entropy is also striking: Figure 3(b) shows that ablating weakening neurons often makes the output distribution flatter; in other words weakening neurons make the output distribution *sharper*. We would expect the opposite: removing information from the residual stream should make it less informative and therefore flatten the output distribution.

## 6.2 CONDITIONAL ABLATIONS

We now further investigate the effect of weakening neurons on entropy. We use **conditional ablations**: We ablate only some activations of each neuron, based on the signs of the corresponding $x_{\text{gate}}$ and $x_{\text{in}}$. Specifically, we consider the following four conditions (using the preprocessing from section 3.2): (i) $x_{\text{gate}} > 0, x_{\text{in}} > 0$, leading to $x_{\text{post}} > 0$; (ii) $x_{\text{gate}} > 0, x_{\text{in}} < 0$, leading to $x_{\text{post}} < 0$; (iii) $x_{\text{gate}} < 0, x_{\text{in}} < 0$, leading to $x_{\text{post}} > 0$; (iv) $x_{\text{gate}} < 0, x_{\text{in}} > 0$, leading to $x_{\text{post}} < 0$.

We find that a large part of the sharpening effect of weakening neurons is due to case (iii): In figure 3(b), case (iii) (bottom left subplot) shows entropy effects similar to those of weakening neurons as a whole, whereas this is much less the case for the other subplots. This is surprising, but also solves the mystery we encountered earlier (i.e., we expected weakening neurons to flatten the distribution, but in reality they often sharpen it).

It is *surprising* for two reasons: First, these negative $x_{\text{gate}}$ activations are relatively rare in weakening neurons (as we will see in section 7). Second, because of the Swish function, **negative gate values** are relatively small (whereas positive values can be arbitrarily large), and it was often assumed they were

only useful for training dynamics (see section 3.1). Our results show for the first time (concurrently with Kong et al. (2025) who focus on a different phenomenon) that negative gate values have a strong effect on model mechanisms (not just training). *This shows that, for mechanistic interpretability research, Swish is not reducible to ReLU.*

This is also *explanatory*: When $x_{gate} < 0$, the usual neuron behavior gets a minus sign in front, so that weakening neurons take on a strengthening behavior. E.g., if a neuron usually detects "minus *again*" ($w_{gate}$) and writes "*again*" ($w_{out}$), in case (iii) it detects "*again*" ($-w_{gate}$) and writes "*again*" ($w_{out}$), which indeed makes the output distribution sharper. We will analyze such a neuron in section 8.

### 6.3 CASE STUDY OF ENTROPY REDUCTION

To understand this phenomenon further, we study a particular text example, namely where the entropy reduction by case (iii) activations of weakening neurons was most extreme (with zero ablation).

The input text is: *Yesterday (21 December) the Government announced a package of support for hospitality and leisure businesses that are losing trade because of the O* and the correct next token is *mic* (as in *Omicron*). The model predicts this next token correctly.

Which tokens have the largest score difference between clean and ablated runs? We find that, in the clean run, *mic* and similar tokens get a massive boost (of up to 12 points) compared to the ablated run, whereas no token gets its score reduced by nearly as much. Thus, at least in this case, the case (iii) activations of weakening neurons sharpen the output distribution by boosting the correct next token.

We further investigate whether any single weakening neuron has a $w_{out}$ similar to *mic*. This is not the case. This suggests that in this case weakening neurons work together (in superposition, cf. Elhage et al., 2022) to achieve the observed effect.

## 7 WEAKENING NEURONS ACTIVATE OFTEN

Our findings from section 6 raise the question of how often weakening neurons activate, i.e., how often their gate value is positive. In fact, Gurnee et al. (2024) found a negative correlation between activation frequency and $\cos(w_{in}, w_{out})$ – but in a GELU model. We now investigate whether a similar phenomenon occurs with gated activation functions. We show the most striking result in figure 4. We show other results in section J and discuss some of them in section G.

Consistent with Gurnee et al. (2024), we find that the many (conditional) strengthening neurons activate very rarely, and (conditional) **weakening neurons activate very often**. In fact, in most layers there is an almost linear negative relationship between $\cos(w_{in}, w_{out})$ and activation frequency: correlations are at least $-0.71$ in all layers except the last two (which have $-0.29$ and $+0.29$).

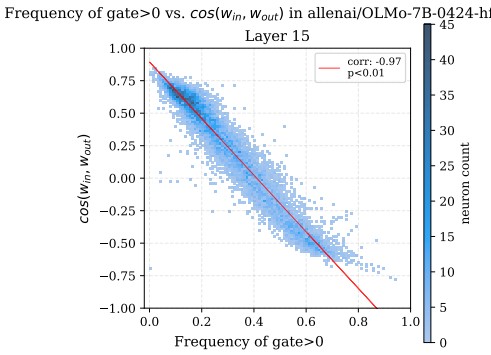

Figure 4: Two-dimensional histogram of activation frequency (x-axis) vs. $\cos(w_{in}, w_{out})$ (y-axis).

It seems that each conditional strengthening neuron is responsible only for a narrow domain, perhaps a specific set of tokens. This result is another indication that weakening neurons have a disproportionately large influence on model behavior. Note however that activation frequencies do not fully explain their effect, since we found that even their negative gate values are influential (section 6).

## 8 CASE STUDY OF A WEAKENING NEURON

We now qualitatively examine two neurons in more detail: a strengthening and a weakening neuron. We will see that weakening neurons can have a quite complex behavior. In section I we detail the

methods (including how we chose the neurons), present this analysis in more detail, and present many more case studies from various RW functionalities.

To analyze the neurons, we combine the RW perspective with two well-established neuron analysis methods: projecting weights to vocabulary space (nostalgebraist, 2020; Geva et al., 2022; Dar et al., 2023; Gurnee et al., 2024; Voita et al., 2024), and finding text examples which strongly activate the neuron (Dalvi et al., 2019; Geva et al., 2021; Nanda, 2022; Voita et al., 2024; Gurnee et al., 2024).

We choose neuron **28.4737** for strengthening and **31.9634** for weakening.[12] Both of them are also a *prediction* neuron in the sense of Gurnee et al. (2024), which indicates that they directly promote a specific set of tokens and are thus likely to be monosemantic.

We can see that **strengthening neuron 28.4737** has a straightforward read-write behavior: It further promotes *review* when the residual stream already indicates that this is the obvious next token.

In contrast, **weakening neuron 31.9634** is much harder to interpret: The weights indicate that this neuron produces "*again*" when the residual stream contains "minus *again*"; but the examples strongly activating the neuron do not have an obvious semantic relationship to *again*.

The most interpretable activations were some weaker positive activations when *again* is a plausible continuation, e.g., on the token ***once*** (as in *once again*). These are cases with negative $x_{\text{gate}}$ values (and also $x_{\text{in}} < 0$, hence positive activations) – a case that we found to be important in section 6.2. In these cases, *again* is already weakly present in the residual stream before the last MLP, and the neuron reinforces *again*. Thus the behavior of this particular weakening neuron is interpretable in the $x_{\text{gate}} < 0$ case, echoing our finding from section 6.2 that this case is surprisingly relevant to model behavior.

These two case studies show that even when the output weights are highly interpretable, strengthening and weakening have a very different overall behavior, and the weakening behavior is much more complex. We think that this is due to the nature of weakening: it inherently involves (an apparent) conflict between the intermediate model prediction and what the neuron promotes.

## 9 CONCLUSION

We have explored a new method for analyzing gated neurons in LLMs: computing the cosine similarities of their weights to understand their read-write functionality. Our method complements prior interpretability approaches and, though quite simple, provides striking new insights into the inner working of LLMs.

We have found that a large share of neurons exhibit strong RW interactions: early-middle layers are dominated by conditional strengthening neurons; weakening neurons are fewer and appear mostly in late layers. This finding is particularly noteworthy since it is universal across models, and is all the more striking since it could be observed with such a simple method.

Focusing on weakening neurons, a relatively small RW class, we have discovered that they have an outsize impact on model behavior, including aspects as different as attribute rate (part of factual recall), and next-token entropy. We have also introduced a new analysis method, conditional ablation, which enables to find out which activations of a neuron are responsible for a given behavior. This method has shown that part of the impact of weakening neurons is due to a mechanism involving negative gate values; we are the first to observe such a mechanism.

Our findings open up new research questions in mechanistic interpretability, and we hope that our study will inspire further investigations. In particular, a better understanding of weakening neurons is crucial for interpreting LLMs overall. Investigating the many conditional strengthening neurons in more detail could also lead to valuable insights. In upcoming work, we plan to investigate the evolution of RW functionalities during model training. Later on, we would also like to go beyond the analysis of single neurons and address questions such as how neurons work together within and across RW classes, or whether a similar analysis also works for SAE latents.

---

[12]The notation is "layer.neuron", with zero-based indexing. The model – still OLMo-7B – has 32 layers, so our weakening neuron is in the final layer.

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

## A  LIMITATIONS

We focus on a *parameter-based* interpretation of *single neurons*. This has the advantage of being simple and efficient, but is also inherently limited in scope. Accordingly, our method is not designed to replace other neuron analysis methods, but to complement them.

The mathematical similarities of weights are insightful, but they should not be taken as one-to-one representations of semantic similarity. We find cases in which close-to-orthogonal vectors represent very similar concepts (double checking, section H).

Finally, while our findings are highly significant and relevant, we do not yet fully understand the reasons behind them.

## B  DEONTOLOGY STATEMENTS

### B.1  LICENSES AND LANGUAGES OF MODELS AND DATA

**Gemma.** To download the model one needs to explicitly accept the terms of use. NLP research is explicitly listed as an intended usage. Primarily English and code Gemma (2024).

**Llama.** Inference code and weights under an ad hoc license. There is also an "Acceptable Use Policy". Our work is well within those terms. Languages mostly include English and programming languages, but also Wikipedia dumps from "bg, ca, cs, da, de, en, es, fr, hr, hu, it, nl, pl, pt, ro, ru, sl, sr, sv, uk" Touvron et al. (2023).

**OLMo and Dolma.** Training and inference code, weights (OLMo), and data (Dolma) under Apache 2.0 license. "The Science of Language Models" is explicitly mentioned as an intended use case. Dolma is quality-filtered and designed to contain only English and programming languages (though we came across some French sentences as well, see table 3) Groeneveld et al. (2024); Soldaini et al. (2024).

**Mistral.** Inference code and weights are released under the Apache 2.0 license, but accessing them requires accepting the terms. Languages are not explicitly mentioned in the paper, but clearly include English and code Jiang et al. (2023).

**Qwen.** Inference code and weights under Apache 2.0 license. Supports "over 29 languages, including Chinese, English, French, Spanish, Portuguese, German, Italian, Russian, Japanese, Korean, Vietnamese, Thai, Arabic, and more" Yang et al. (2024).

**Yi.** Inference code and weights under Apache 2.0 license. Trained on English and Chinese 01.AI et al. (2025).

### B.2  COMPUTATIONAL COMPLEXITY

All our experiments can be run on a single NVIDIA RTX A6000 (48GB). We use TransformerLens Nanda & Bloom (2022). A colleague kindly provided us with a version that also supports OLMo.

The main analysis, computing the weight cosines, needs less than a minute per model.

Other parts were more expensive:

- For the ablations (section 6), each run on Dolma used took approximately 8 hours.

- For the activation-based analysis in section 8, we needed a single run of $\approx 25$ h to store the max/min activating examples for all neurons, and then $\approx 45$ s per neuron ($\approx 5$ min) to recompute its activations on the relevant texts and visualize them.

- Another expensive part is computing the randomness regions based on mismatched cosines (sections 4.3 and E). The time complexity is $O(n^2)$ in the number of neurons per layer, since we have to consider every pair of neurons. Since however we found that this baseline is hardly different from the more "naive" Gaussian one, we suggest that future work could just leave out this step.

- Finally, our weight processing makes model loading last about a minute. A possible solution in future work would be to save the preprocessed weights.

### B.3  LLM USE

We used LLM assistants to help with programming.

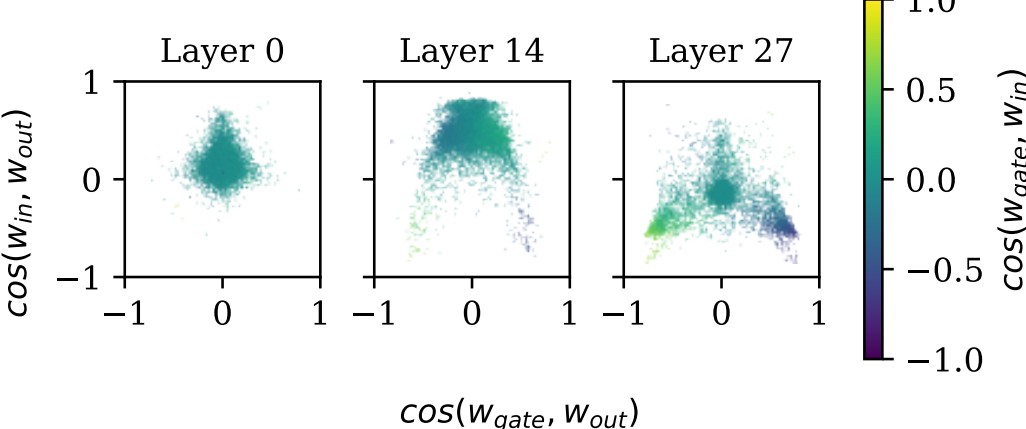

Figure 5: Equivalent of figure 52, but *without* weight processing. (We don't include the randomness regions.)

## C   WEIGHT PREPROCESSING

In this section we argue for our weight preprocessing step described in section 3.2: multiplying $w_{\text{in}}$ and $w_{\text{out}}$ by the sign of $\cos(w_{\text{gate}}, w_{\text{in}})$.

First of all, this processing **does not affect model behavior**: In equation (2), if we replace $w_{\text{in}}$ by $-w_{\text{in}}$ and $w_{\text{out}}$ by $-w_{\text{out}}$, the two minus signs cancel out. We call this the **symmetry property** of gated activation functions.

We find neurons slightly **easier to interpret** after applying this weight preprocessing step, for the following reasons:

- The two reading weight vectors, $w_{\text{gate}}$ and $w_{\text{in}}$, now always have a non-negative cosine similarity, i.e., they do not point in opposite directions. This makes it easier to reason about what causes a neuron to activate (there are less minus signs to worry about). This is especially relevant for the case studies (section 8).

- In scatter plots like figure 2, neurons that belong together are in the same area of the plot. Consider any dot in figure 2 (which was made *with* the weight processing), for example on the bottom left of the subplot (i.e. a weakening neuron). If we undo the weight processing, $\cos(w_{\text{gate}}, w_{\text{out}})$ (say $-0.8$) may be randomly replaced by its opposite (say $+0.8$), in which case the same neuron would now appear on the bottom *right*. Accordingly, in the equivalent figure *without* weight processing (figure 5), we see two distinct clusters of weakening neurons, at the bottom left and bottom right, while these clusters are not different in terms of neuron behavior.

- Similarly, for the conditional ablations introduced in section 6.2, the four cases correspond to real distinctions. Without weight preprocessing, equivalent cases would be more complicated to define (e.g. "gate+_post+" would be defined as "$x_{\text{gate}} > 0$ and $x_{\text{post}}$ has the same sign as $\cos(w_{\text{gate}}, w_{\text{in}})$").

## D   FROM NEURONS TO LINEAR COMBINATIONS

In section 4.1 we claimed that neuron-based findings can carry over to linear combinations of neurons, at least to some extent.

We will give the argument for one example: a combination of two prototypical strengthening neurons (let's call them $n^{(1)}$ and $n^{(2)}$)). Let their weights be $w_{\text{gate}}^{(1)} = w_{\text{in}}^{(1)} = w_{\text{out}}^{(1)} := w^{(1)}$ and

$\boldsymbol{w}_{\text{gate}}^{(2)} = \boldsymbol{w}_{\text{in}}^{(2)} = \boldsymbol{w}_{\text{out}}^{(2)} := \boldsymbol{w}^{(2)}$. We consider the multi-neuron feature $n^{(1)} + n^{(2)}$ (defined by $n^{(1)}$ and $n^{(2)}$ activating together with the same strengths).

This feature will activate when $\boldsymbol{x}_{\text{norm}}$ contains both $\boldsymbol{w}^{(1)}$ and $\boldsymbol{w}^{(2)}$, which implies containing $\boldsymbol{w}^{(1)} + \boldsymbol{w}^{(2)}$ (though the converse does not hold). When activating, it will write a positive multiple to $\boldsymbol{w}^{(1)} + \boldsymbol{w}^{(2)}$ to the residual stream. Thus this linear combination of strengthening neurons has a strengthening-like behavior itself.

Similar arguments can be made for any pair of neurons of the same RW class, and also for more than two neurons.

In particular, in a layer containing many conditional strengthening neurons (like early-middle layers of all models), a sparse linear combination of neurons is reasonably likely to consist of conditional strengthening neurons, in which case it will display a behavior similar to conditional strengthening.

## E  RANDOM BASELINES

Here we describe our two baselines: random initialization and mismatched cosines.

In a randomly initialized model, all cosine similarities would be very close to zero: In $n$ dimensions, absolute cosine similarities behave like $1/\sqrt{n}$ (Vershynin, 2025, p. 68). More precisely, the cosines follow a beta distribution with parameters $(d_{\text{model}} - 1)/2, (d_{\text{model}} - 1)/2$, rescaled to the range $[-1, 1]$.[13] Taking, e.g., $d_{\text{model}} = 4096$ (as e.g. in OLMo-7B), we get a 95% randomness range of approximately $[-0.03, 0.03]$. This is empirically confirmed on the first training checkpoint of OLMo-7B-0424 (figure 6).

Inspired by work on outlier dimensions in the *activations* of Transformers (Ethayarajh, 2019; Kovaleva et al., 2021; Timkey & van Schijndel, 2021; Dettmers et al., 2022; Sun et al., 2024), we suspected that a similar phenomenon might be at work in the *weights*, making cosine similarities artificially high. To account for this possibility, we construct a second baseline specific to each model layer: We compute all the (e.g.) $\cos(\boldsymbol{w}_{\text{in}}, \boldsymbol{w}_{\text{out}})$ of a layer, even if the two weights belong to different neurons. If a cosine similarity is higher than most of these mismatched cosines, it is likely not due to an outlier dimension common to all neurons of the layer, but reflects something specific to this neuron.

## F  ABLATION EXPERIMENTS

### F.1  HYPOTHESES AND CHOICE OF METRICS

We originally had two hypotheses (which turned out to be wrong, see section 6):

- We hypothesized that *conditional strengthening* neurons might contribute to *subject enrichment* Geva et al. (2023), a crucial step of factual recall that involves MLPs writing appropriate attributes for the given subject. Both phenomena occur in roughly the same layers, and similar $\boldsymbol{w}_{\text{in}}$ and $\boldsymbol{w}_{\text{out}}$ could correspond to related concepts.
- We expected that *weakening* neurons would make the output distribution flatter, i.e. *increase the entropy*. This could happen by reducing the probability of high-ranking tokens (weakening directions corresponding to tokens) or by increasing the probability of very low-ranking tokens (weakening directions corresponding to negations of tokens).

This is why we tested the two metrics of *attribute rate* (a proxy of subject enrichment) and *entropy*. We additionally considered the *loss*, and, following Gurnee et al., 2024 analysis of entropy neurons, *rank* of the correct token and *scale* of the final hidden state.

### F.2  NUMBER OF NEURONS TO ABLATE

In preliminary experiments, we tried ablating 24 or 243 neurons in each run, which is the number of strengthening or weakening neurons in OLMo, respectively. Ablating 24 neurons (of any class) did

---

[13]https://stats.stackexchange.com/questions/85916/distribution-of-scalar-products-of-two-random-unit-vectors-in-d-dimensions

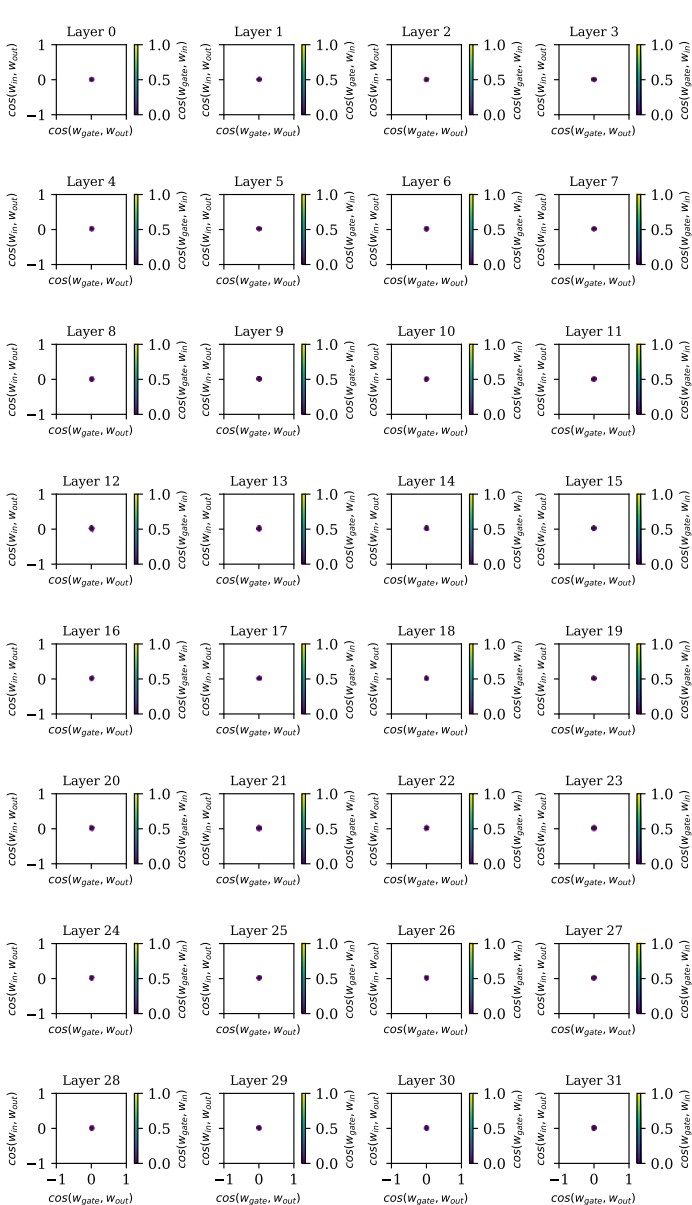

Figure 6: Equivalent of figure 2 for the randomly initialized OLMo-7B model (training checkpoint 0). Whatever doesn't look like this, is significant.

not have a clear impact on any metric (including loss), so we sticked with 243, which has a visible impact on the loss, but does not break the model altogether. For this reason we exclude strengthening neurons from most of our experiments, since there are only 24 of them.

### F.3 DETAILS ON ATTRIBUTE RATE

Our investigation of attribute rate closely follows Geva et al. (2023). It requires a dataset of subject-attribute mappings that we didn't have access to. In order to replicate this dataset, we closely followed the procedure described in their paper, which assumes attributes are tokens that appear in the same Wikipedia paragraph as the subject (excluding stopwords). We used the Wikipedia dump from October 20, 2021, instead of October 13, since there is an official dump made at this date.[14] To improve replicability, we publish our complete code as well as our subject-attribute dataset.

### F.4 MEAN ABLATION

#### F.4.1 METHOD

**Computing the means.** We pre-computed the mean activation of every neuron on the same 20M token subset of Dolma that we also used for the actual ablation experiment.

**Conditional ablations.** For conditional ablations, we replace the neuron activation by the mean value it would have in the corresponding case (not the mean activation of the neuron overall). For example, in the case "gate-_post+":

- we replace the activation ($x_{\text{post}}$) only when the condition "gate-_post+" is fulfilled, i.e. when $x_{\text{gate}} < 0$ and $x_{\text{post}} > 0$ – this is just the definition of conditional ablation;
- the value that we replace it with is the mean value of $x_{\text{post}}$ *across the cases in which $x_{gate} < 0$ and $x_{post} > 0$*.

#### F.4.2 RESULTS

We show the results in figures 10 to 13, 25 to 32 and 36 to 39.

Regarding **attributes rate**, we find again that non-weakening neurons have no effect (figures 11 to 13 and 25). Mean-ablating weakening neurons has a somewhat inconsistent effect (figure 10): In middle layers the intervention reduces attributes rate (though less starkly than with zero ablation), suggesting that these neurons play a role in subject enrichment. In some late layers however (especially layer 29), the intervention *increases* attributes rate, suggesting that the neurons help avoiding an explosion of the attributes rate at this computation stage. (This would make sense: at the end of the forward pass the model wants to predict the next token, not some attribute of the subject.)

Regarding **entropy**, it remains the case that ablating weakening neurons has the biggest effect (figure 29). (In both zero and ablation, strengthening neurons also often reduce entropy, but this is expected. See figures 17 and 25) In particular, there is still a substantial number of cases in which weakening neurons *reduce* entropy by about 10 nats. However, entropy changes are now more evenly distributed: there is also a substantial (only slightly smaller) amount of cases in which weakening neurons *increase* entropy, which aligns better with their expected behavior. **Conditional ablations** (figure 36) partially recover both effects with the case "gate-_post+" (as in zero ablation), but also with the case "gate+_post-" (contrary to zero ablation).

Mean ablation thus recovers effects of weakening neurons that would go unnoticed when just using zero ablation. We hypothesize that these additional effects happen when activations are relatively close to zero but far away from their mean. This remains to be tested in future work.

## G ACTIVATION FREQUENCIES

This section extends section 7.

Complete results are in table 5 and figures 7 to 9.

---

[14]A list of dumps by date is available at `https://archive.org/search?query=subject`.

Table 2: Overview of prediction/suppression neurons chosen for case studies in section 8

| Neuron | RW category | $\cos(\boldsymbol{w}_{\text{gate}}, \boldsymbol{w}_{\text{in}})$ | $\cos(\boldsymbol{w}_{\text{gate}}, \boldsymbol{w}_{\text{out}})$ | $\cos(\boldsymbol{w}_{\text{in}}, \boldsymbol{w}_{\text{out}})$ |
|---|---|---|---|---|
| 28.4737 | strengthening | 0.5290 | 0.5048 | 0.7060 |
| 28.9766 | conditional strengthening | 0.4764 | 0.4119 | 0.5982 |
| 31.9634 | weakening | -0.7164 | 0.7218 | -0.8542 |
| 29.10900 | conditional weakening | 0.4988 | -0.4992 | -0.5775 |
| 30.10972 | proportional change | -0.4543 | 0.5814 | -0.4182 |
| 29.4180 | orthogonal output | -0.0272 | -0.4057 | 0.0669 |

The last layer displays a different pattern than the rest (last subplot in figure 7). Here the correlation is positive ($+0.29$), and we can distinguish two clusters of neurons: One cluster has a medium-negative $\cos(\boldsymbol{w}_{\text{in}}, \boldsymbol{w}_{\text{out}})$ (around $-0.3$) and activates very rarely; another one is much more spread out (both in terms of $\cos(\boldsymbol{w}_{\text{in}}, \boldsymbol{w}_{\text{out}})$ and activation frequency), centers at a weaker negative cosine similarity ($-0.1$ to $-0.2$) and activates a bit more than half of the time. The presence of these two clusters leads to the slightly positive correlation. Comparing with the other plots suggests that the first cluster mostly corresponds to weakening neurons and atypical proportional change neurons.

We do not find such striking patterns with gate-out or gate-in similarities.

## H  DOUBLE CHECKING

In our case studies, we observe that many neurons have the property of **double checking**: The two reading weight vectors ($\boldsymbol{w}_{\text{gate}}$ and $\boldsymbol{w}_{\text{in}}$) are approximately orthogonal, but still intuitively represent the same concept.

We characterize double checking as follows: The sets of meaningful vectors *most similar* to $\boldsymbol{w}_{\text{gate}}$ and $\boldsymbol{w}_{\text{in}}$ have a high overlap. More formally, let $U = \{u_0, ..., u_{d_{\text{vocab}}}\}$ be the set of unembedding vectors; then

$$\arg\max_{u \in U} \cos(u, \boldsymbol{w}_{\text{gate}}) \approx \arg\max_{u \in U} \cos(u, \boldsymbol{w}_{\text{in}}).$$

This phenomenon is *possible* because random vectors in high dimensions are "lone stars" (Vershynin, 2025, p. 68). If this is the case for the unembedding vectors, it is plausible that we can find $\boldsymbol{w}_{\text{gate}}, \boldsymbol{w}_{\text{in}}$ that are reasonably similar to a $u_i$ but not to any other $u_j$. These $\boldsymbol{w}_{\text{gate}}, \boldsymbol{w}_{\text{in}}$ can even be (approximately) orthogonal to each other, as in the following three-dimensional toy example: $u_1 = (1, 0, 0), u_2 = (0, 1, 0), \boldsymbol{w}_{\text{gate}} = (1, 0, 1), \boldsymbol{w}_{\text{in}} = (1, 0, -1)$.

However the phenomenon is *unlikely* to occur in random vectors, and hence is a significant finding: If choosing $\boldsymbol{w}_{\text{gate}}, \boldsymbol{w}_{\text{in}}$ randomly, we would expect them to be approximately orthogonal to *all* unembedding vectors; and even if both were somewhat similar to an unembedding, we certainly wouldn't expect it to be the same unembedding for both.

We would also not naively expect this phenomenon in a trained network: If the role of both $\boldsymbol{w}_{\text{gate}}$ and $\boldsymbol{w}_{\text{in}}$ is to detect a concept (e.g. a token prediction) represented by a vector $u$, then we would get the best performance with $\boldsymbol{w}_{\text{gate}} = \boldsymbol{w}_{\text{in}} = u$, i.e., $\boldsymbol{w}_{\text{gate}}, \boldsymbol{w}_{\text{in}}$ would not be orthogonal.

Double checking is therefore likely to be a useful feature for the model. We hypothesize that this is because it shrinks the region in model space that activates the neuron positively. If (say) $\boldsymbol{w}_{\text{in}} = \boldsymbol{w}_{\text{gate}} = (1, 0)$, the neuron activates whenever the (normalized) residual input $x$ satisfies $x \cdot (1, 0) > 0$; this happens on the whole half-space $x_1 > 0$. If however $\boldsymbol{w}_{\text{gate}} = (1, 0)$ and $\boldsymbol{w}_{\text{in}} = (0, 1)$, the neuron activates positively only in the first quadrant ($x_1, x_2 > 0$).

This behavior thus enables more precise concept detection. This may explain why conditional neurons are more frequent than their unconditional counterparts.

Table 3: Description of the weight vectors of the selected *prediction* neurons, by top tokens or similarity to $w_{out}$. The question mark, ?, signals unknown unicode characters. The last column presents the (shortened) text samples on which the respective neuron activates most strongly (positively or negatively).

| Neuron, RW class | $w_{gate}$ | | $w_{in}$ | | $w_{out}$ | Top activations |
|---|---|---|---|---|---|---|
| 28.4737 strengthening | $\approx w_{out}$ | | $\approx w_{out}$ | | pos: *review Review* | pos (13.75): *Download EBOOK [...] Description of the book [...] \n -> Reviews* neg (-2.25): *The answer's at the bottom of **this** -> post* |
| 28.9766 conditional strengthening | pos: *well well* | neg: *far high* | $\approx w_{out}$ | | pos: *well well* | pos (18.63): *Could have saved myself some time. **Oh** -> , well* neg (-3.66): *Seek to understand them **more** -> fully* |
| 31.9634 weakening | $\approx -w_{out}$ | | $\approx -w_{out}$ | | pos: *again Again* | pos (3.48): *the areas of the doorjamb where the **door** -> often* neg (-5.12): *jumping off the roof of his Los Angeles apartment building. -> Meanwhile* |
| 29.10900 conditional weakening | pos: *today nowadays* | neg: *these these* | $\approx -w_{out}$ | | pos: *these These* | pos (12.79): *social media tools change and come and go at the drop of a **hat** -> .* neg (-2.18): *la couleur de sa robe **et** -> le* |
| 30.10972 proportional change | $\approx w_{out}$ | | pos: *when when* | neg: *timing dates* | neg: *when when* | pos (6.14): *puts you on multiple webpages **at** -> as soon as* neg (-2.67): *Take pleasure in the rest of the new year. -> You* |
| 29.4180 orthogonal output | pos: *here therein* | neg: *there we* | pos: ? | neg: *here in* | neg: *there there* | pos (2.31): *without any consideration being issued or paid **there** -> for* neg (-14.41): *here **or** -> there* |

Table 4: Description of the weight vectors of the selected *prototypical* neurons, by top tokens or similarity to $\boldsymbol{w}_{\text{out}}$. The question mark, ?, signals unknown unicode characters. The last column presents the (shortened) text samples on which the respective neuron activates most strongly (positively or negatively).

| Neuron, RW class | $\boldsymbol{w}_{\text{gate}}$ | | $\boldsymbol{w}_{\text{in}}$ | | $\boldsymbol{w}_{\text{out}}$ | | Top activations |
|---|---|---|---|---|---|---|---|
| 25.9997 strengthening | $\approx \boldsymbol{w}_{\text{out}}$ | | $\approx \boldsymbol{w}_{\text{out}}$ | | pos: *S* *S* | neg: *Chocolate* *Cour* | pos (10.45): *when Stannis gives his opinion on **Janos** -> Slynt* 
 neg (-0.72): *said owner Hieu Than, -> who* |
| 5.10602 conditional strengthening | pos: *t* *as* | neg: *deep* *hum* | $\approx \boldsymbol{w}_{\text{out}}$ | | pos: *as* *t* | neg: *ating* *their* | pos (1.56): *a big-time rocker, playing **aren** -> as* 
 neg (-0.68): *workers **don** -> 't* |
| 31.7117 weakening | $\approx -\boldsymbol{w}_{\text{out}}$ | | $\approx -\boldsymbol{w}_{\text{out}}$ | | pos: *by* *by* | neg: *ani* *iw* | pos (7.27): *deux projets de décision figu**rant** -> dans* 
 neg (-5.23): *Take pleasure in the rest of the new year. -> You* |
| 23.6543 conditional weakening | pos: *the* *a* | neg: *ham* *aden* | $\approx -\boldsymbol{w}_{\text{out}}$ | | pos: *Op* *AB* | neg: *rom* *c* | pos (1.06): *High-value and sub-high-**value** -> areas* 
 neg (-0.99): *:basis **and** -> :sub-category* |
| 25.7415 proportional change | $\approx \boldsymbol{w}_{\text{out}}$ | | pos: *berry* *rod* | neg: *a* *the* | pos: *?* *Hart* | neg: *Nine* *jin* | pos (2.36): *180 °C ( -> 350 °F)* 
 neg (-1.55): *// @Component({ \n // -> selector* |

## I  CASE STUDIES

### I.1  NEURON CHOICE

We used two different methods to find interesting neurons:

**First**, we selected among *prediction neurons* in the sense of Gurnee et al. (2024). These are defined as neurons whose $\cos(W_U, \boldsymbol{w}_{\text{out}})$ has a high kurtosis; in other words, they boost predictions of a small set of tokens while leaving other token scores virtually unchanged. Specifically, from each discrete RW class we chose the neuron with the highest kurtosis. This first method guarantees finding interpretable neurons in terms of output behavior, though not necessarily an interpretable *overall* behavior. See table 2 for an overview of neurons chosen by this method.

A downside is that prediction neurons tend to appear in later layers only. Therefore this neuron choice does not help understand what happens in early layers, especially why there are so many conditional strengthening neurons. We therefore also use a **second** method: We just select the most prototypical neuron from each class. For example, for conditional strengthening, we take the neuron with the highest $\cos(\boldsymbol{w}_{\text{in}}, \boldsymbol{w}_{\text{out}})$ among those neurons whose $\cos(\boldsymbol{w}_{\text{gate}}, \boldsymbol{w}_{\text{out}})$ is within the randomness range (sections 4.3 and E). This method led to choosing the neurons 5.10602 (conditional strengthening), 23.6543 (conditional weakening), 25.7415 (proportional change), 25.9997 (strengthening), 31.7117 (weakening).

### I.2  METHODS

Additionally to our RW analysis, we use two well-established neuron analysis methods:

First, we project neuron weights to vocabulary space with the unembedding matrix $W_U$ and inspect high-scoring tokens.

Second, we find text examples on which the neurons are strongly activated (positively or negatively). For each neuron we save the 16 strongest positive and negative activations, respectively.

### I.3    DETAILED ANALYSIS OF WEAKENING NEURON 31.9634

Here we say a bit more about the neuron analyzed in section 8.

Judging by the weights, we would predict the following: The neuron activates positively when the residual stream contains the "minus *again*" direction, and then weakens that direction by writing "plus *again*". The neuron activates negatively when the residual stream contains information both for and against predicting *again*, and then weakens the *again* direction. Given that $w_{\text{gate}}$ and $w_{\text{in}}$ are highly similar ($\cos(w_{\text{gate}}, w_{\text{in}}) = 0.7164$), we would expect that it is easier for the neuron to activate positively (with $x_{\text{gate}}$ and $x_{\text{in}}$ of the same sign).

When actually recording activations of the neuron, we get a more complex picture: First of all, the neuron often activates negatively. Strong negative activations are often on punctuation, and the actual next token is often *meanwhile* or *instead* (and not *again*). On the positive side, the strongest activations do not have any obvious semantic relationship to *again*. We also observed weaker positive activations when *again* is a plausible continuation, e.g., on the token **once** (as in *once again*). These are cases with negative $x_{\text{gate}}$ values (and also $x_{\text{in}} < 0$, hence positive activations) – a case that we found to be important in section 6.2. In these cases, *again* is already weakly present in the residual stream before the last MLP, and the neuron reinforces *again*.

Thus the behavior of this particular weakening neuron is interpretable in the $x_{\text{gate}} < 0$ case, echoing our finding from section 6.2 that this case is surprisingly relevant to model behavior. The $x_{\text{gate}} > 0$ case is less interpretable for this particular neuron, even though this case is more frequent and can lead to stronger activations. Nevertheless we have some hypotheses for the strong activations as well: For strong positive activations (which showed no clear pattern), we hypothesize that sometimes the residual stream ends up near "minus *again*" for semantically unrelated reasons (there are many more possible concepts than dimensions, so the corresponding directions cannot be fully orthogonal; see Elhage et al., 2022); in these cases the neuron would reduce the unjustified presence of this "minus *again*" direction. With strong negative activations (where the next token was often *meanwhile* or *instead*), the neuron may ensure only these tokens are predicted, and not the relatively similar *again*.

### I.4    RESULTS AND ANALYSIS FOR PREDICTION NEURONS

See table 3.

**Strengthening neuron 28.4737**[15] predicts *review* (and related tokens) if activated positively, which happens if *review* is already present in the residual stream. The maximally positive activations are in standard contexts that continue with *review* or similar, such as the newline after the description of an e-book (the next paragraph often is the beginning of a review).

**Conditional strengthening neuron 28.9766's** RW functionality concerns *well* and similar tokens. 28.9766 promotes them if activated positively, which happens when both $w_{\text{gate}}$ and $w_{\text{in}}$ indicate that *well* is represented in the residual stream. This is a case of double checking. The maximally positive activation in our sample occurs on **Oh**, in a context in which *Oh, well* makes sense (and is the actual continuation).

**Weakening neuron 31.9634.** See section I.3.

**Conditional weakening neuron 29.10900.** Gate and linear input weight vectors act as two independent ways of checking that *these* is not present in the residual stream (i.e., a case of double checking). At the same time, they check for predictions like *today, nowadays*. When such predictions are present, the neuron promotes *these*. This is a plausible choice in these cases because of the expression *these days*. An example is *social media tools change and come and go at the drop of a **hat***. (This sentence talks about a characteristic of current times, so *these days* would indeed be a plausible continuation.)

**Proportional change neuron 30.10972** predicts the token *when* if activated negatively. This happens if *when* is absent from the residual stream (gate condition) and is proportional to the presence of time-related tokens (-$w_{\text{in}}$). An example for a large negative activation is *puts you on multiple webpages*

---

[15]The notation is "layer.neuron", with zero-based indexing.

*at*.[16] Conversely, if *when* is absent, and time-related tokens are absent too, the neuron activates positively and suppresses *when* further.

**Orthogonal output neuron 29.4180** predicts *there* (positive activation) if the residual stream contains a component that we interpret as "complement of place expected" (e.g., *here*, *therein*). Both $w_{\text{gate}}$ and $w_{\text{in}}$ check for (different aspects of) this component being present, another case of double checking. The largest positive activation is on *here or*.

Overall, these neurons all promote a specific set of tokens (we chose them that way), but under very different circumstances. The (conditional) strengthening neurons are the most straightforward to interpret, because their input and output clearly correspond to the same concept. In contrast, weakening neurons inherently involve (an apparent) conflict between the intermediate model prediction and what the neuron promotes.

### I.5    RESULTS AND ANALYSIS FOR PROTOTYPICAL EXAMPLES

See table 4.

**Strengthening neuron 25.9997.**

**Conditional strengthening neuron 5.10602** activates on those tokens that often start negated auxiliary verbs: *don, aren, won, didn*. Correspondingly, the top token of $W_U w_{\text{gate}}$ is *t* (but interestingly not an apostrophe). On the other hand, $w_{\text{in}}$ detects alternative predictions: *ate, ating* etc. (as in *donate*) lead to a negative activation, and *as* (as in *arenas*) leads to a positive activation. Correspondingly the strongest positive activations are on the *aren* of *arenas* (but the strongest negative activations are not always in a *donate* context, perhaps because both *don't* and *donate* can appear in the same slots). These alternative predictions are then strengthened by $w_{\text{out}}$.

**Weakening neuron 31.7117.**

**Conditional weakening neurons 23.6543.**

**Proportional change neuron 25.7415.**

### I.6    MORE CASE STUDIES

These are various neurons that popped out to us as possibly interesting, for not very systematic reasons, for example because they strongly activated on a specific named entity. All of them are in OLMo-7B. We present them by RW class. For most of these case studies we did only a quick and dirty weight-based analysis. In some cases we also tried $W_E$ (input embeddings) instead of $W_U$ (unembeddings) for the logit-lens style analysis.

#### I.6.1    CONDITIONAL STRENGTHENING NEURONS

**0.1480**: $w_{\text{gate}}, -w_{\text{in}}, -w_{\text{out}}$ all have tokens similar to *box* (when using $W_E$). Activates on *Xbox*.

**4.1940**: *country* appears in $w_{\text{in}}$ among many other things. When using $W_E$, *Philippines* and *Manila* appear in $w_{\text{out}}$. Activates on *Philippines*.

**4.3720**: gate seems country/government related. When using $W_E$, we find $w_{\text{out}}, w_{\text{gate}}$ contain some country names. Activates on *Denmark*.

**4.4801**: *Muhammad* appears in the gate vector. Activates on *Muhammad*.

**4.5772**: predicts *ian* as in *Egyptian*. When using $W_E$, all three weight vectors contain *Egypt*. Activates on *Egypt*.

---

[16]The actual sentence ends with *as soon as* and comes from a now-dead webpage. We also found one occurrence of *at when* in what seems to be a paraphrase of the same text, on https://www.docdroid.net/RgxdG5s/fantastic-tips-for-bloggers-of-all-amountsoystcpdf-pdf . We suspect that both texts are machine-generated paraphrases of an original text containing *at once* (*when* and *as soon as* can be synonyms of *once* in other contexts), and that the model has (also) seen a paraphrased version with *at when*. In fact many of the largest negative activations are on *at* in contexts calling for *at once*.

**4.6517** has a very Ireland (or Celtic nations) related gate vector. The interpretations of the other two weights are less obvious, but *Irish* and *Dublin* appear in $w_{\text{in}}$ among many other things, and *UK* and *London* appear in $-w_{\text{out}}$ (Ireland is emphatically *not* in the UK!) When using $W_E$, *Ireland* appears among the top tokens of all three weight vectors. Activates on *Ireland*.

**4.6799**: When using $W_E$, *Vietnam* is among the tokens corresponding to $-w_{\text{out}}$. Activates on *Vietnam*

**4.7667**: all three weights related to consoles in different ways. Activates on *Xbox*

**4.9983**: $w_{\text{out}}$ is related to electronic devices, $w_{\text{in}}$ either electronic devices or sports (surfing may belong to both), $w_{\text{gate}}$ is also mostly related to electronic devices. When using $W_E$, we find $w_{\text{out}}$ contains *iPhone* as a top token. Activates on *iPhone*.

**4.10859**: When using $W_E$, we find $w_{\text{gate}}, w_{\text{out}}$ include *Thailand* as a top token, $w_{\text{out}}$ additionally *Buddha, Buddhist*. Activates on *Thailand*.

**4.10882**: When using $W_E$, we find $-w_{\text{out}}$ contains *Italy*, $-w_{\text{in}}, w_{\text{gate}}$ additionally contain *Rome*. Activates on *Italy*.

**4.10995**: *Boston* appears in gate and *Massachusetts* in $-w_{\text{in}}$. When using $W_E$, we find $-w_{\text{out}}, w_{\text{gate}}$ contain *Massachusetts* and *Boston*, $-w_{\text{in}}$ contains *Boston*. Activates on *Massachusetts*.

**22.2589**: $w_{\text{gate}}$ and $-w_{\text{in}}$ recognize tokens like *Islam, Muhammad* and others related to the Arabo-Islamic world. The same goes for $-w_{\text{out}}$ (as it is similar to $w_{\text{in}}$). Activates on *Muhammad*.

**24.4880**: For all three weight vectors the first four tokens (but not more) are Philippine-related (even though the gate vector is actually not very similar to the others). The gate vector also reacts to other geographical names, which *may* have in common that they are associated with non-"white" (Black, Asian or Latin) people in the US sense (*Singapore, Malaysian, Nigerian, Seoul, Pacific, Kerala, Bangkok*, but also *(Los) Angeles* and *Bronx*). Activates on *Philippines*.

**24.6771**: $w_{\text{gate}}, -w_{\text{in}}, -w_{\text{out}}$ all correspond to capitalized first names. Activates on *Muhammad*.

**25.2723**: Some tokens associated with $w_{\text{in}}$ and $w_{\text{out}}$ are possible completions for *th* (*th-ousand, th-ought, th-orn*. When using $W_E$, in all three weights there are a few *th* tokens, but also with *ph* and similar. Activates on *Thailand*.

**25.10496**: $-w_{\text{in}}, -w_{\text{out}}$ correspond to tokens starting with *v* (upper or lower case, with or without preceding space). $w_{\text{gate}}$ on the other hand seems to react to appropriate endings for tokens starting in *v*: *vol-atility, v-antage, v-intage, vel-ocity, V-ancouver*. When using $W_E$, we also find all three weight-vectors are very *v*-heavy. Activates on *Vietnam*.

### I.6.2 WEAKENING NEURONS

**30.9996**: Downgrades weird tokens if present / promotes frequent English stopwords if absent. Also an attention deactivation neuron for 15 heads in layer 31.

### I.6.3 PROPORTIONAL CHANGE NEURONS

**25.7032**: Some tokens associated with $w_{\text{gate}}$ and $w_{\text{out}}$ are possible completions for *x* or *ex* (*X-avier, x-yz, ex-cel, ex-ercise*. When using $W_E$, both *x* and *box* (with variants) appear in all three weight vectors. Activates on *Xbox*.

**25.8607**: All three vectors correspond to tokens related to cities. Moreover, $-w_{\text{out}}$ seems to correspond to non-city places, such as national governments or villages. $w_{\text{in}}$ is actually not that similar to $w_{\text{gate}}, w_{\text{out}}$ (in terms of cosine similarities), but all three correspond to city-related tokens. When using $W_E$, in all three weights there are a few city-related tokens. Activates on *Paris*. We may think of the two input directions as two largely independent ways of checking that "it's about a city" (this is a recurring phenomenon that we describe in section H). When the gate activates but the linear input does not confirm it's about a city, the output promotes closely related but non-city interpretations (for example *Paris* actually refers to the French government in some contexts).

**29.8118**: Partition neuron, highest variance of all proportional change neurons. Also an attention deactivation neuron for 4 heads (0,2,11,15) in layer 30.

**31.5490**: Activates on *Muhammad*. $w_{\text{gate}}$ reacts to various Asian names and Asian-sounding subwords, $w_{\text{in}}$ to surnames as opposed to other English words starting with space and uppercase letter. $w_{\text{out}}$ corresponds to more Asian stuff (mostly subwords) as opposed to English surnames.

**31.6275**: Mostly promotes two-letter tokens (no preceding space, typically uppercase). $-w_{\text{in}}$ typically lowercase single letters. $-w_{\text{gate}}$ mostly lowercase two-letter tokens. "If no lowercase two-letter tokens, promote uppercase two-letter tokens proportionally to absence of lowercase single letters" ?

**31.8342**: This is an *-ot-* neuron: $w_{\text{gate}}$ and $w_{\text{out}}$ correspond to *-o(t)-* suffixes, $-w_{\text{in}}$ to various *-ot-* stuff. Judging by the weight similarities, we expect that $w_{\text{out}}$ is typically activated negatively: downgrade *-o(t)-* suffixes if present in the residual stream. Activates on *Egypt*.

### I.6.4 ORTHOGONAL OUTPUT NEURONS

**0.1758**: When using $W_E$, all three weight vectors' top tokens are famous web sites, including *YouTube*. Activates on *YouTube*.

**0.3338**: When using $W_E$, we find especially $w_{\text{gate}}$ and $-w_{\text{in}}$, but also $-w_{\text{out}}$ are similar to smartphone-related tokens. Activates on *iPhone*.

**0.3872**: When using $W_E$, we find especially $w_{\text{gate}}$, but also $-w_{\text{in}}$ and $-w_{\text{out}}$ correspond to city names. Activates on *Paris*.

**0.7829**: When using $W_E$, we find $w_{\text{in}}, w_{\text{out}}$ and to a lesser extent $w_{\text{gate}}$ correspond in large part to software names. Activates on *iTunes*.

**0.7966**: When using $W_E$, the weight vectors mostly correspond to tokens starting with *th*. Activates on *Thor*.

**29.2568**: $w_{\text{out}}$ Asian (Thai?) sounding syllables vs. (Asian) geographic names in English and other stuff; $w_{\text{in}}$ reacts to Thailand and Asian (geography) stuff as opposed to (mostly) US stuff; $w_{\text{gate}}$ pretty much the same. Activates on *Thailand*.

**29.3327**: $w_{\text{gate}}$ mostly reacts to city names (*Paris* being the most important one), $-w_{\text{in}}$ countries and cities, especially in continental Europe (*France* and *Paris* on top) as opposed to stuff related to the former British Empire. Relevant is $-w_{\text{out}}$ which corresponds to pieces of geographical names and especially rivers in France (*Se-ine, Rh-one / Rh-ine, Mar-ne, Mos-elle... Norm-andie, Nancy, commun...*). $w_{\text{gate}}$ and $-w_{\text{in}}$ also react to *river(s)*. Activates on *Paris*.

**29.4101**: $w_{\text{gate}}$ and $w_{\text{in}}$ react to *YouTube* (top token!), $w_{\text{out}}$ downgrades it (almost bottom token) and promotes *subscrib\*, views, channels* etc. Activates on *YouTube*.

**29.6417**: Downgrades *recording* and similar. $w_{\text{gate}}$ and $w_{\text{in}}$ are also similar and involve *iTunes*. Activates on *iTunes*.

**29.9734**: $w_{\text{gate}}$ reacts to the East in a broad sense as opposed to the West (*Iran, Kaz-akhstan, Kash-mir, Ukraine...*), $w_{\text{in}}$ mostly to male first names without preceding space. $w_{\text{out}}$ seems to produce word pieces that could begin a foreign name. Activates on *Muhammad*.

**30.2667**: $w_{\text{gate}}$ reacts to suffixes (for adjectives derived from place names) like *en, ian, ians*, basically the same for $w_{\text{in}}$ and $w_{\text{out}}$. Activates on *Muhammad*.

**30.3143**: $w_{\text{gate}}$ reacts to words related to entities that are authoritative for various reasons (*officials, authorities, according, researchers, spokesman, investigators...*). $-w_{\text{in}}$ reacts to uncertainty (*reportedly, according... allegedly... accused*). $-w_{\text{out}}$ is again *police, authorities, officials, court* but with no preceding space. Activates on *Philippines*. What authorities and uncertainty have to do with the Philippines is unclear.

**30.3883**: $w_{\text{gate}}$ and $-w_{\text{in}}$ react to *Virginia* and *Afghanistan*, among others (in the case of $w_{\text{gate}}$: as opposed to other geographical names with no preceding space associated with the South and the sea); $-w_{\text{out}}$ is activated and promotes all variants of *af* (and *ghan*) but downgrades Virginia etc. Activates on *Afghanistan*.

**30.4577**: Seems to be related to rugby: $w_{\text{gate}}$ and slightly less obviously $w_{\text{in}}$ react to rugby-related tokens (*midfielder, quarterback...*); $w_{\text{out}}$ promotes different tokens that upon reflection could be related to rugby as well. Activates on *Ireland*.

**30.5372**: Promotes *natural* and related, downgrades *inst* tokens. $w_{\text{in}}$ reacts to *wildlife* etc. as opposed to *institute* etc, $w_{\text{gate}}$ reacts to *institute* as opposed to *natural*. Activates on *Massachusetts* (in which situation it promotes *Institute*, which makes sense because of MIT).

**30.8535**: $-w_{\text{out}}$ is *one* in all variants, $w_{\text{gate}}$ too, $w_{\text{in}}$ splits *one, ones* and the equivalent Chinese characters, on the positive side, from *One, 1, ONE* on the negative side (and many other things on both sides). Activates on *Xbox*. Presumably this happens because *One* is a possible prediction (*Xbox One*), and presumably the output reinforces that.

**31.2135**: orthogonal output, on the conditional strengthening side (weak conditional strengthening, one of the neurons on the vertical axis). $w_{\text{gate}}$ reacts to single letters or symbols as opposed to some English content words without preceding space; $w_{\text{in}}$ and $w_{\text{out}}$ mostly Chinese or Japanese characters as opposed to some Latin diacritics and other weird stuff. Language choice? "If it's not English and single letters are floating around, make sure to choose the right language / character set."

**31.10424**: $w_{\text{gate}}, -w_{\text{in}}, w_{\text{out}}$ correspond to *score* in the top tokens, which is downgraded if present. Activates on *Paris*. No idea what's happening here.

## J MORE PLOTS

These final figures show additional results:

- Table 5 and figures 7 to 9 show more results on activation frequencies: by discrete classes, on all layers, and plotted against $\cos(w_{\text{gate}}, w_{\text{in}})$ or $\cos(w_{\text{gate}}, w_{\text{out}})$.
- Figures 14 to 16 show that ablating 243 neurons from other classes than weakening does not have any effect on attribute rate.
- Figures 17 to 24 show the effect of various ablations on entropy, loss, rank and scale.
- From figure 40, we show our analyses of RW functionalities by layer (section 5) for all the models we investigated.

Regarding the last point, we note a few additional patterns that appear only in some of these models:

- In Yi and the OLMo models, the prevalence of conditional strengthening neurons starts even earlier, at the very first layer. A particularly interesting example is Yi: In layer 0 an enormous 68% of all neurons are conditional strengthening, then almost none, then there is a second wave around layers 11-17 (out of 32) which have around 25% of conditional strengthening neurons each.
- In some models, especially the OLMo ones, there is a non-negligible number of conditional weakening neurons. They tend to appear in middle-to-late layers, shortly after the conditional strengthening wave. The clearest example is OLMo-1B, with a peak of 1418 conditional weakening neurons out of 8192 (17%) in layer 9 out of 16.

The following patterns could be random, but still show that the model has *not* learned something:

- For almost all neurons the cosine similarities are still clearly below 1 (the dots do not fill out the edges in figure 2). This echoes and extends Gurnee et al. (2024)'s findings that in GPT2 the IO cosine similarity is approximately bounded by $\pm 0.8$. In other words, we almost never get the *prototypical* cases of conditional strengthening / weakening etc., as defined in section 4. This might be an effect of randomness (strong cosine similarities are less likely), but could also suggest that even input manipulator neurons add some novel information to the residual stream.
- We also observe that for the vast majority of neurons, $\cos(w_{\text{gate}}, w_{\text{in}}) \approx 0$: This can be seen in the boxplots in the appendix, as well as the purple color in figure 2. Thus most neurons operate on two input directions in the residual stream (not a single one), resulting in higher expressivity and more complex semantics. If not random, this could be related to double checking; see section H.

Table 5: Activation frequencies by RW class. The second column represents random neurons taken from the same layers.

|  | true | baseline |
|---|---|---|
| strengthening | 0.265 | 0.480 |
| atypical strengthening | 0.163 | 0.276 |
| conditional strengthening | 0.132 | 0.370 |
| atypical conditional strengthening | 0.100 | 0.219 |
| proportional change | 0.368 | 0.296 |
| atypical proportional change | 0.344 | 0.338 |
| orthogonal output | 0.373 | 0.236 |
| weakening | 0.691 | 0.384 |
| atypical weakening | 0.727 | 0.353 |
| conditional weakening | 0.708 | 0.301 |
| atypical conditional weakening | 0.665 | 0.469 |

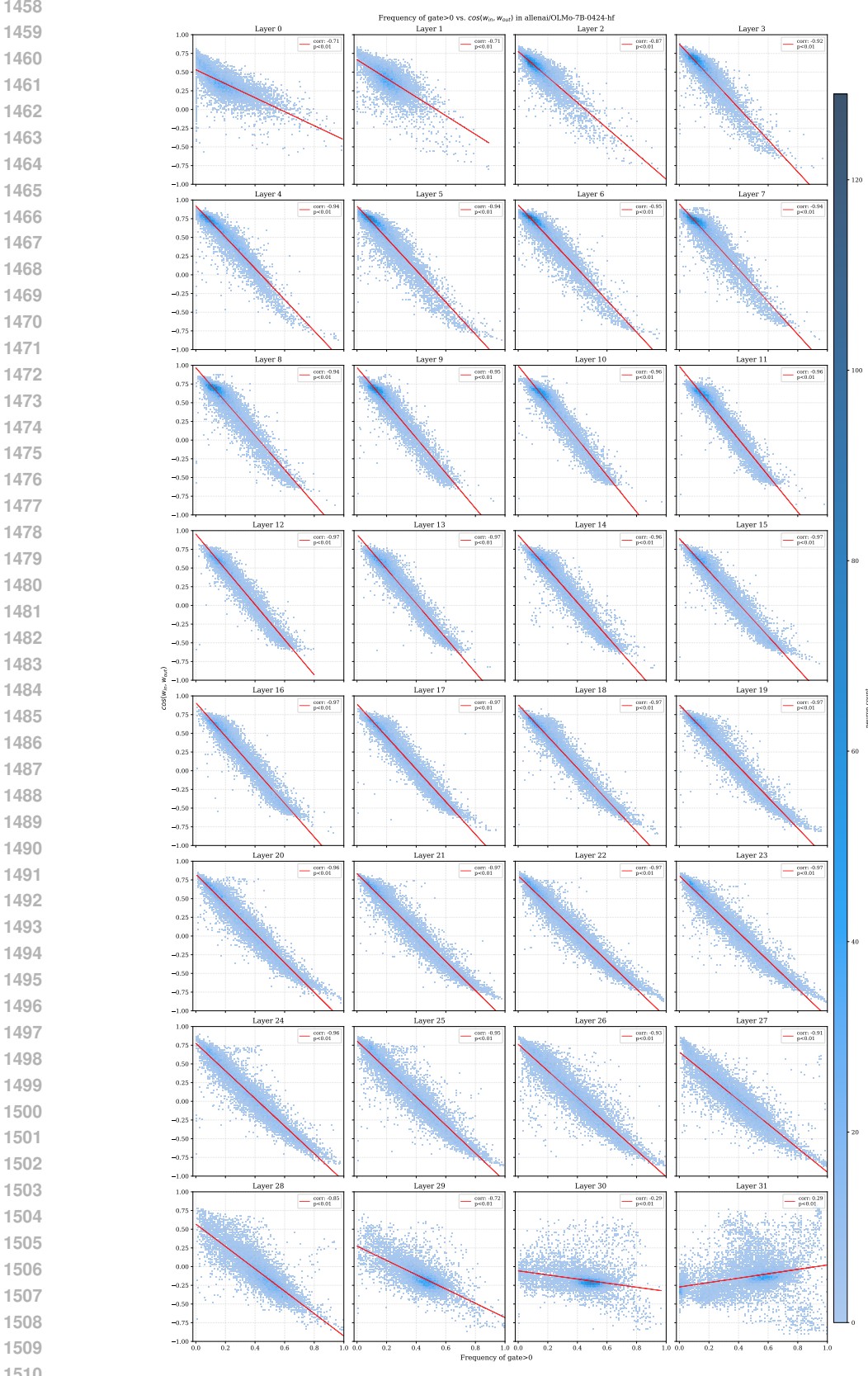

Figure 7: Like figure 4 but for all layers.

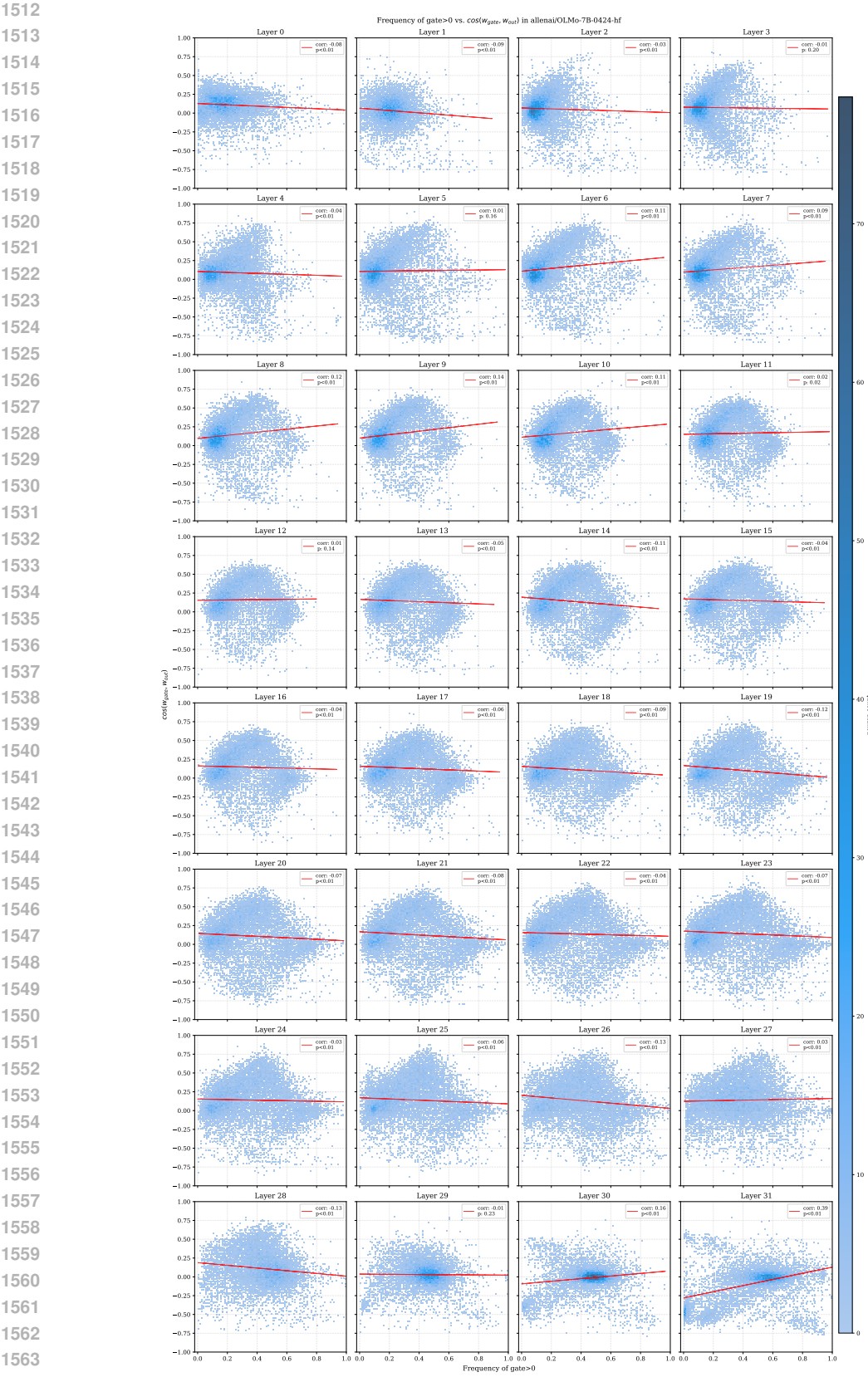

Figure 8: Frequency of gate>0 vs. $|\cos(\boldsymbol{w}_{\text{gate}}, \boldsymbol{w}_{\text{out}})|$ in OLMo-7B.

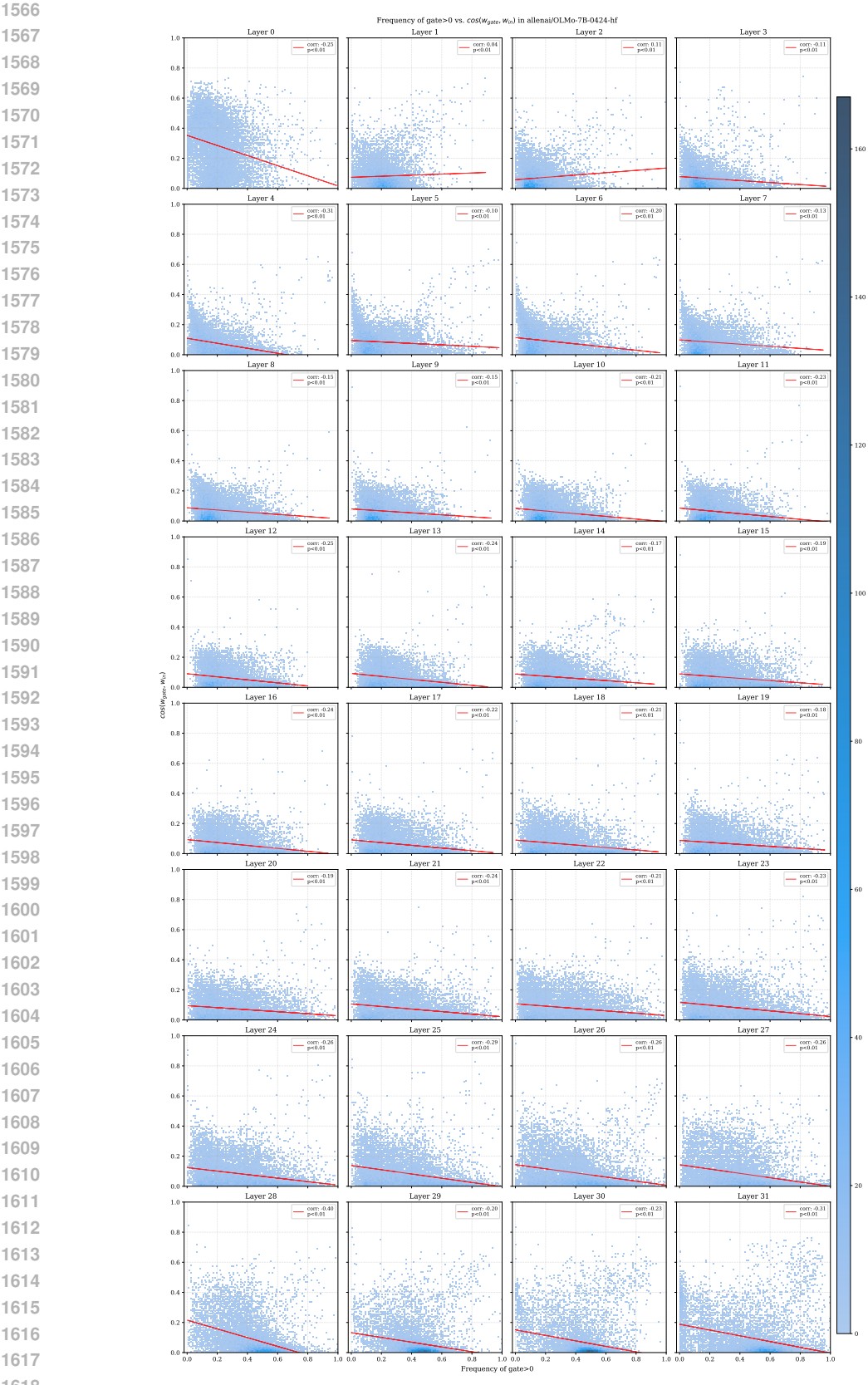

Figure 9: Frequency of gate>0 vs. $\cos(\boldsymbol{w}_{\text{gate}}, \boldsymbol{w}_{\text{in}})$ in OLMo-7B.

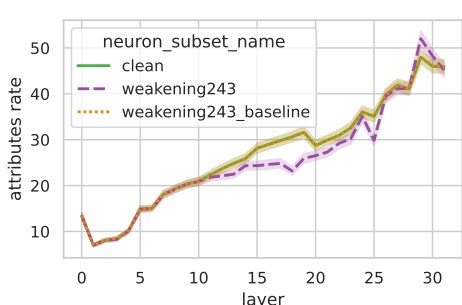

Figure 10: Attribute rate when mean-ablating all 243 weakening neurons.

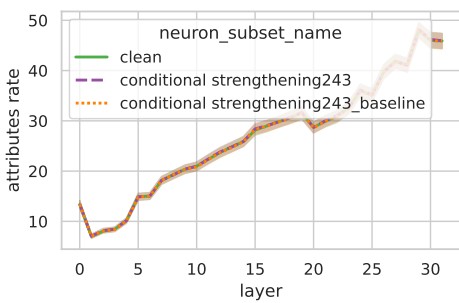

Figure 11: Attribute rate when mean-ablating 243 conditional strengthening neurons.

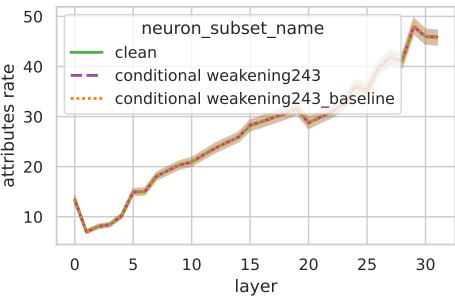

Figure 12: Attribute rate when mean-ablating 243 conditional weakening neurons.

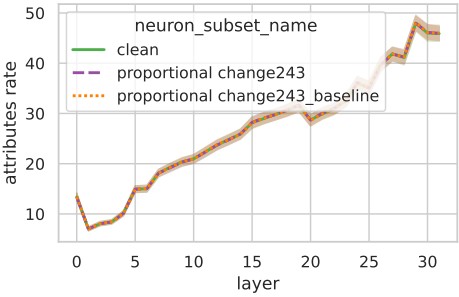

Figure 13: Attribute rate when mean-ablating 243 proportional change neurons.

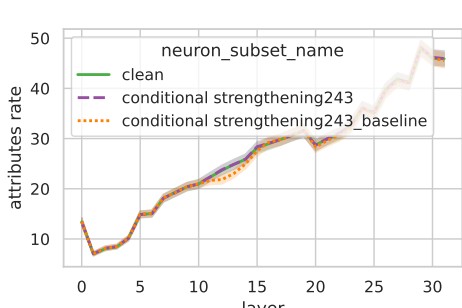

Figure 14: Attribute rate when zero-ablating 243 conditional strengthening neurons.

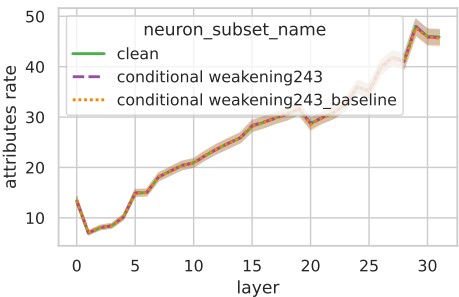

Figure 15: Attribute rate when zero-ablating 243 conditional weakening neurons.

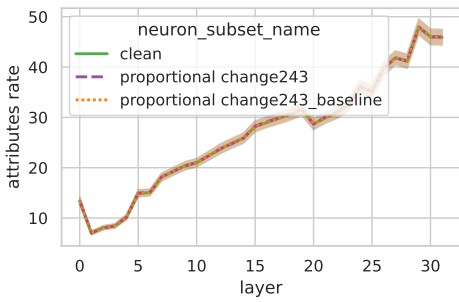

Figure 16: Attribute rate when zero-ablating 243 proportional change neurons.

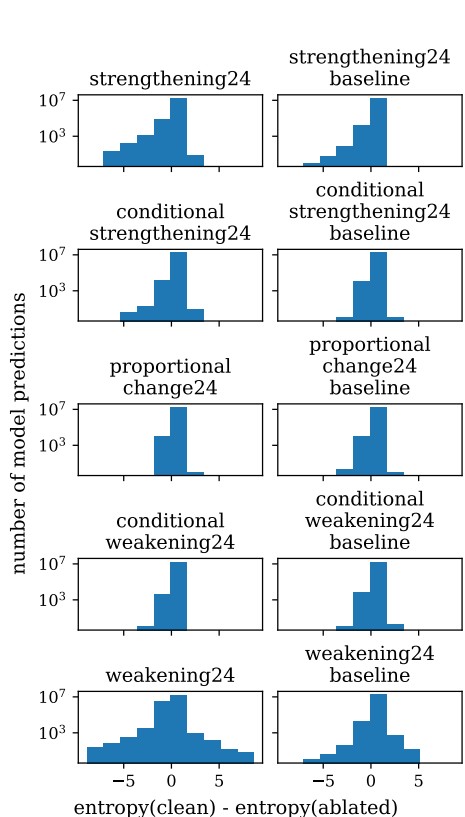

Figure 17: Effect on entropy when zero-ablating 24 neurons from various RW classes.

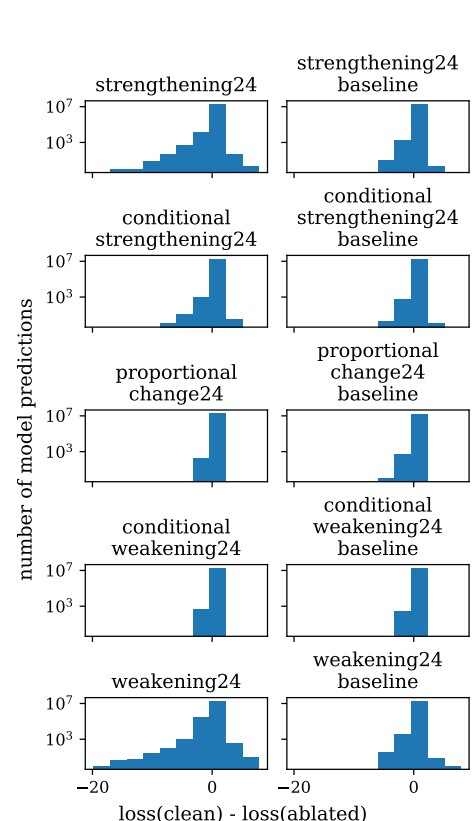

Figure 18: Effect on loss when zero-ablating 24 neurons from various RW classes.

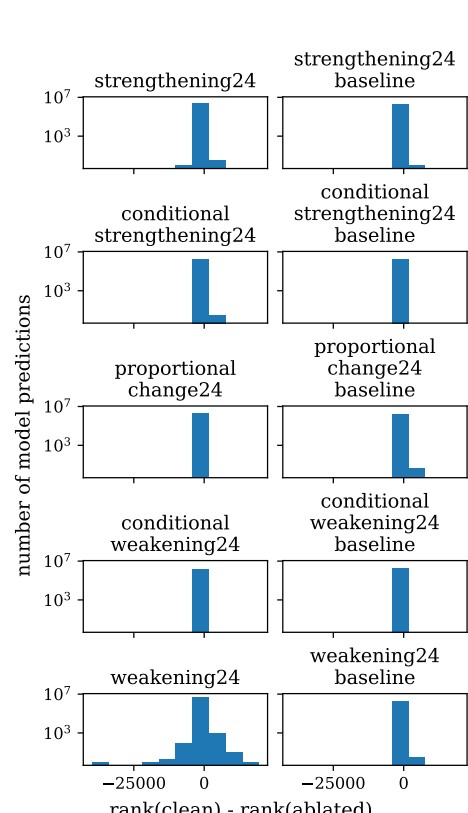

Figure 19: Effect on correct token rank when zero-ablating 24 neurons from various RW classes.

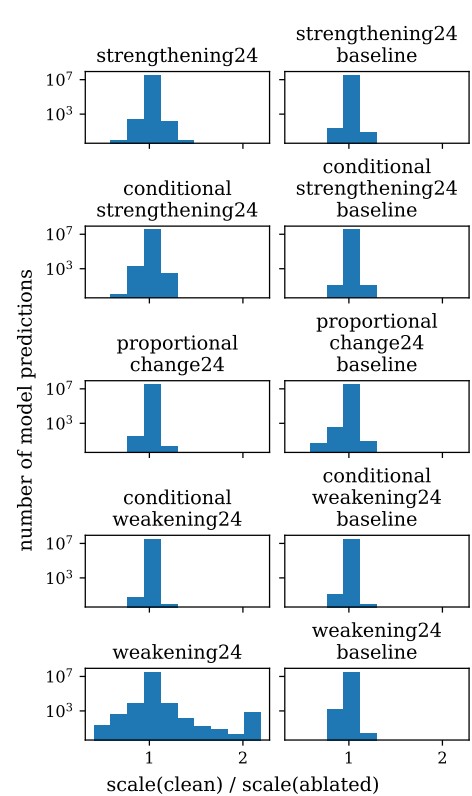

Figure 20: Effect on scale of last hidden state when zero-ablating 24 neurons from various RW classes.

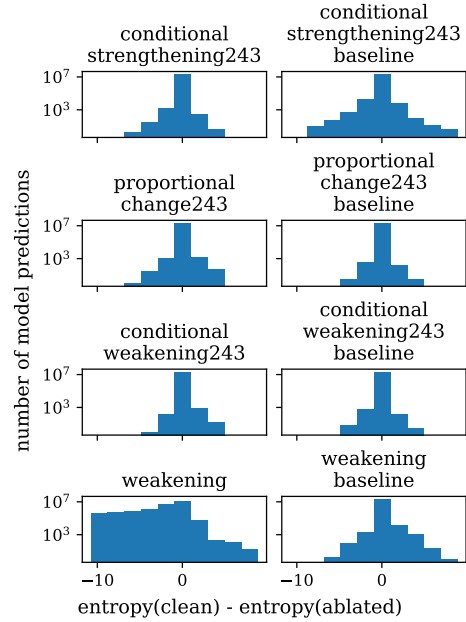

Figure 21: Effect on entropy when zero-ablating 243 neurons from various RW classes.

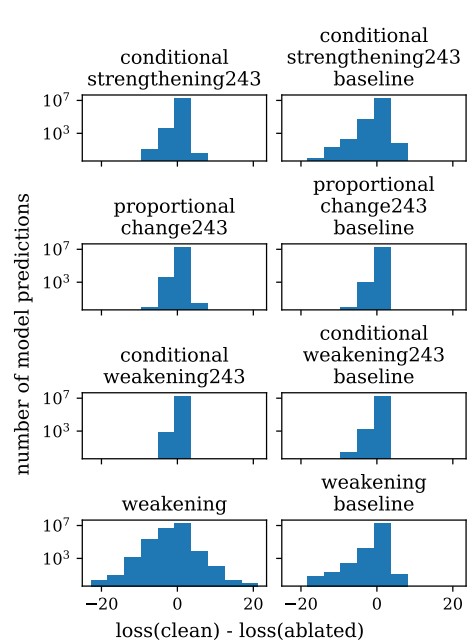

Figure 22: Effect on loss when zero-ablating 243 neurons from various RW classes.

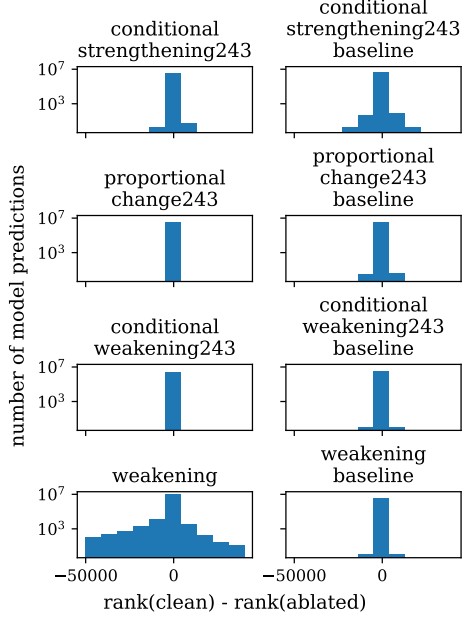

Figure 23: Effect on correct token rank when zero-ablating 243 neurons from various RW classes.

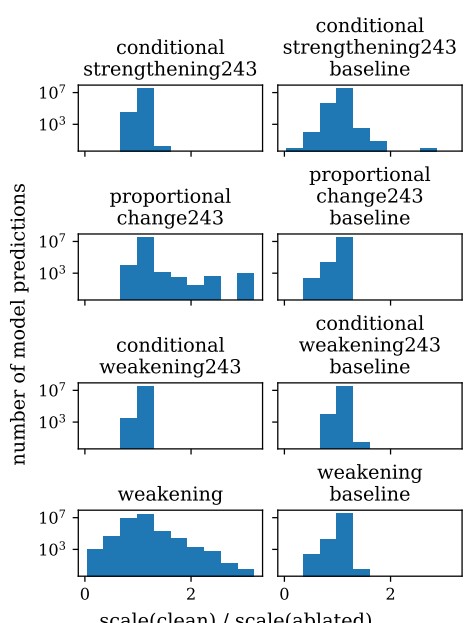

Figure 24: Effect on scale of last hidden state when zero-ablating 243 neurons from various RW classes.

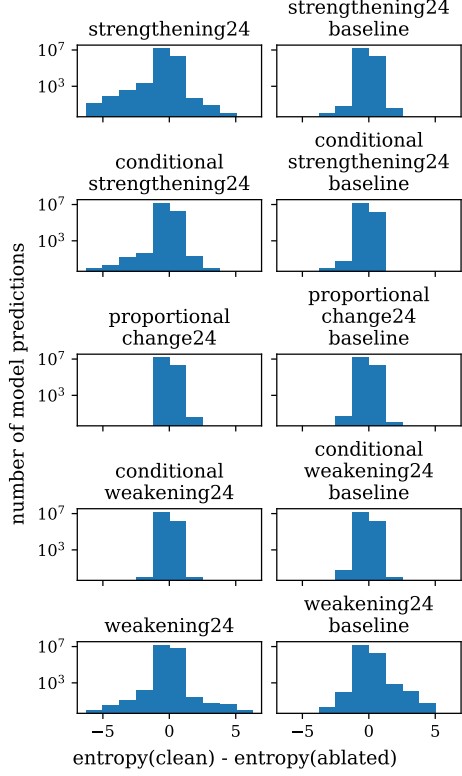

Figure 25: Effect on entropy when mean-ablating 24 neurons from various RW classes.

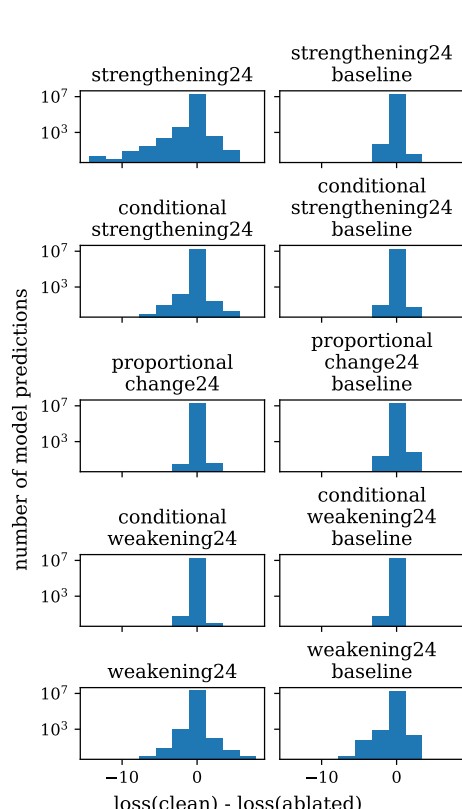

Figure 26: Effect on loss when mean-ablating 24 neurons from various RW classes.

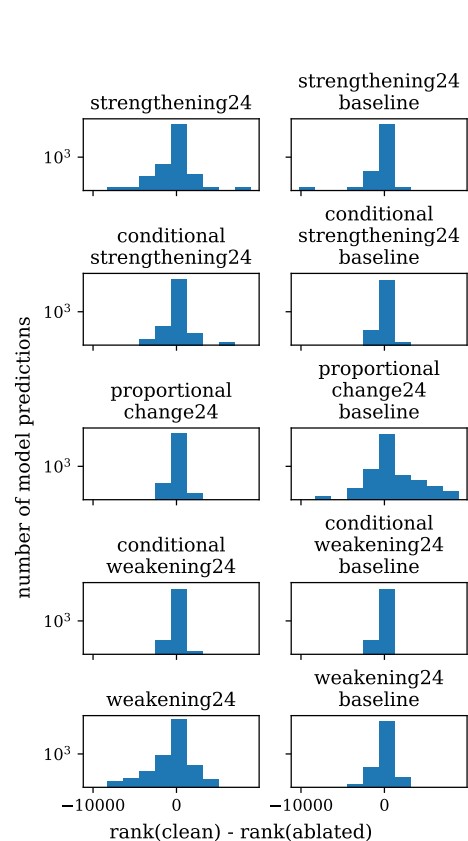

Figure 27: Effect on correct token rank when mean-ablating 24 neurons from various RW classes.

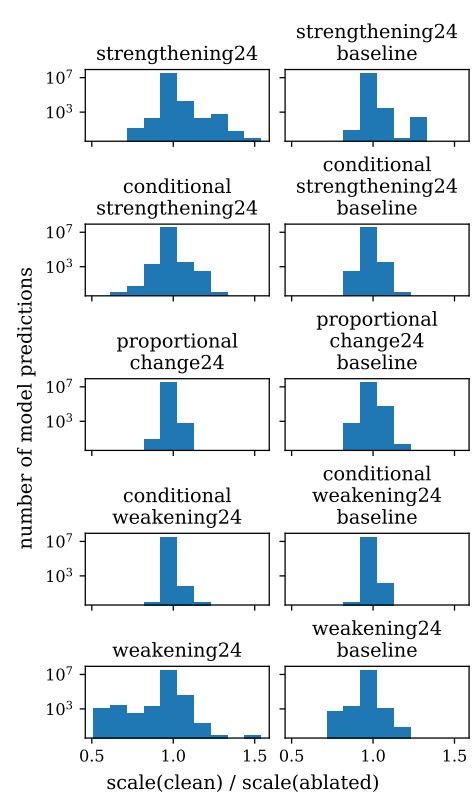

Figure 28: Effect on scale of last hidden state when mean-ablating 24 neurons from various RW classes.

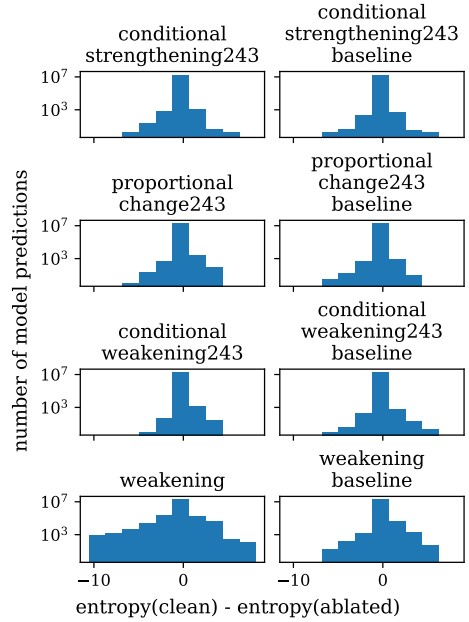

Figure 29: Effect on entropy when mean-ablating 243 neurons from various RW classes.

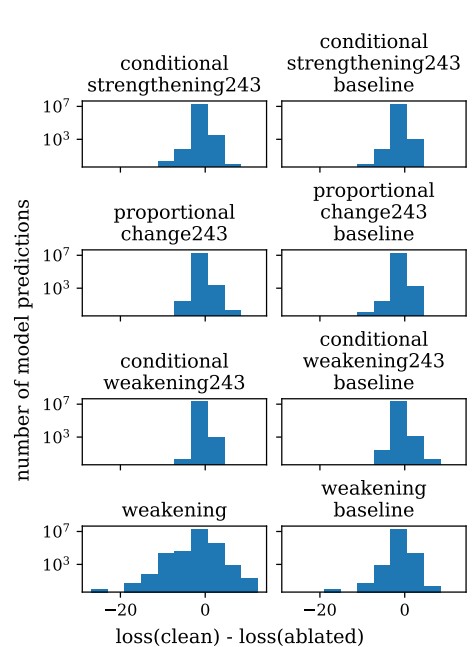

Figure 30: Effect on loss when mean-ablating 243 neurons from various RW classes.

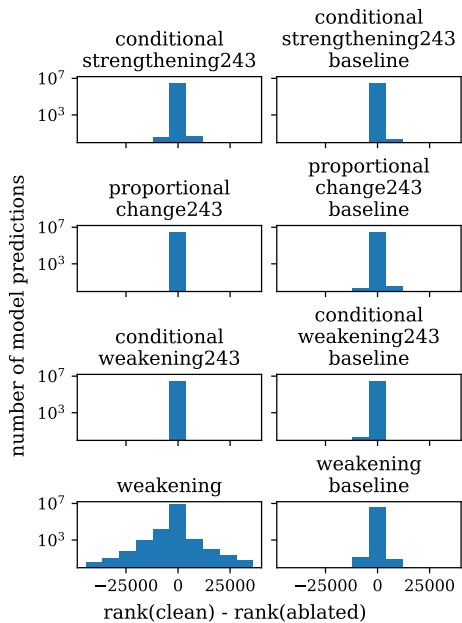

Figure 31: Effect on correct token rank when mean-ablating 243 neurons from various RW classes.

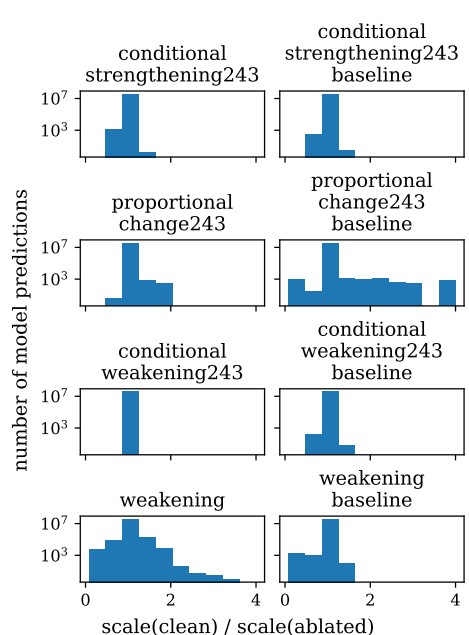

Figure 32: Effect on scale of last hidden state when mean-ablating 243 neurons from various RW classes.

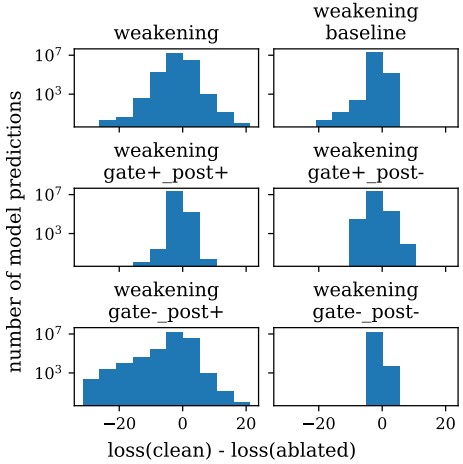

Figure 33: Effect on loss of conditional zero-ablations of weakening neurons.

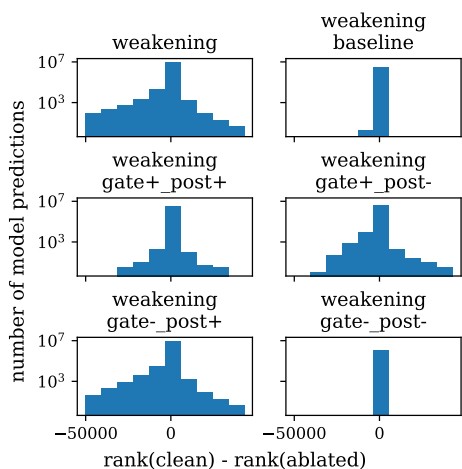

Figure 34: Effect on rank of conditional zero-ablations of weakening neurons.

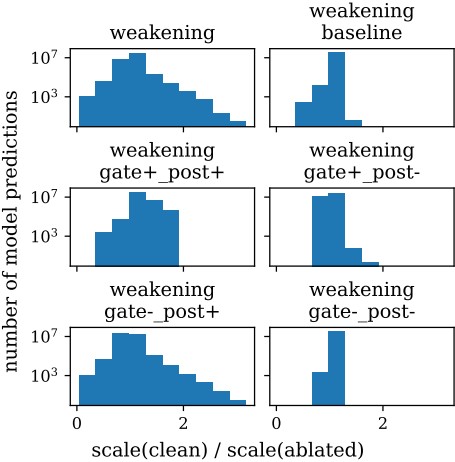

Figure 35: Effect on scale of conditional zero-ablations of weakening neurons.

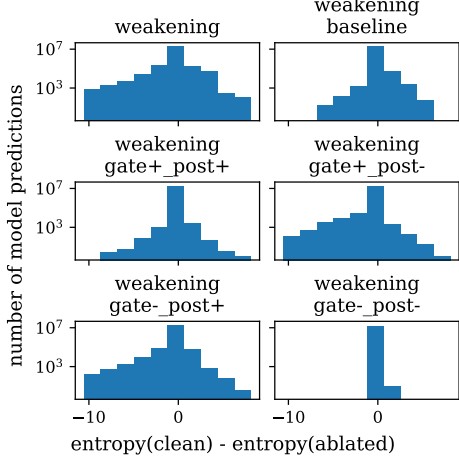

Figure 36: Effect on entropy of conditional mean-ablations of weakening neurons.

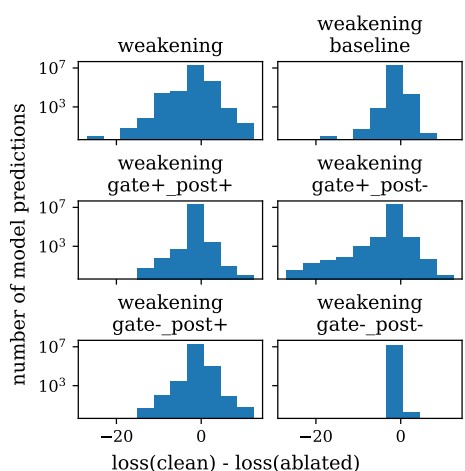

Figure 37: Effect on loss of conditional mean-ablations of weakening neurons.

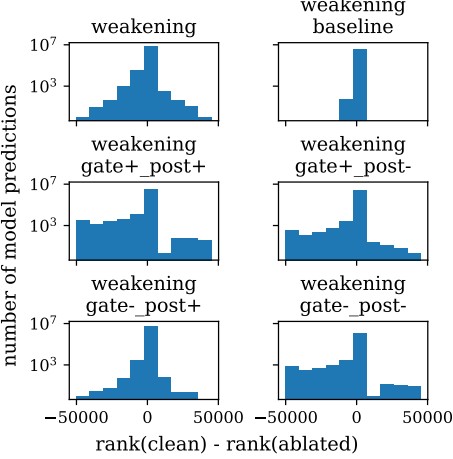

Figure 38: Effect on rank of conditional mean-ablations of weakening neurons.

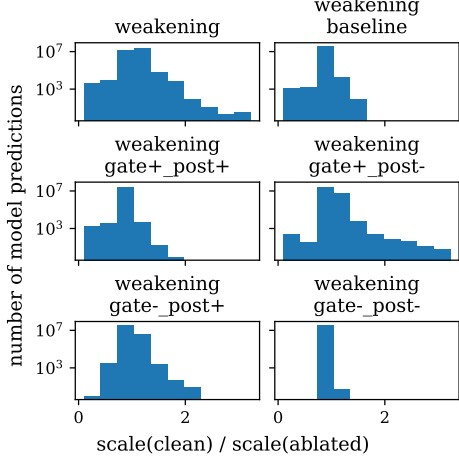

Figure 39: Effect on scale of conditional mean-ablations of weakening neurons.

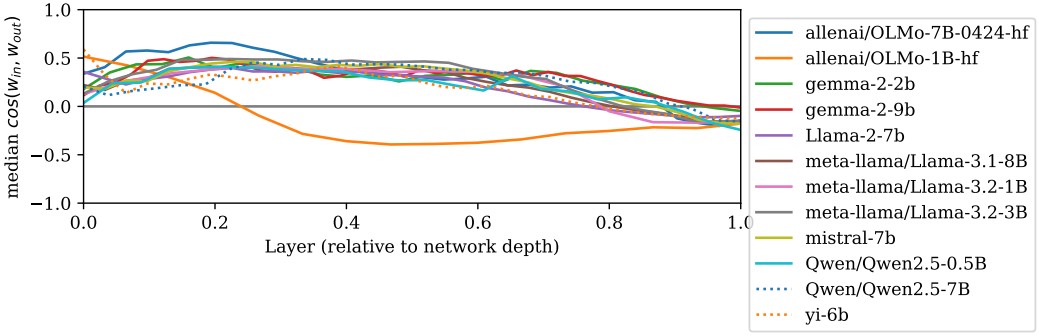

Figure 40: Median of $\cos(\boldsymbol{w}_{\text{in}}, \boldsymbol{w}_{\text{out}})$ by layer (x-axis) for all 12 models investigated. Unlike figure 1(a) we also include the models of 1B parameters and below. All models follow the same general pattern, but OLMo-1B switches to negative values earlier than the others.

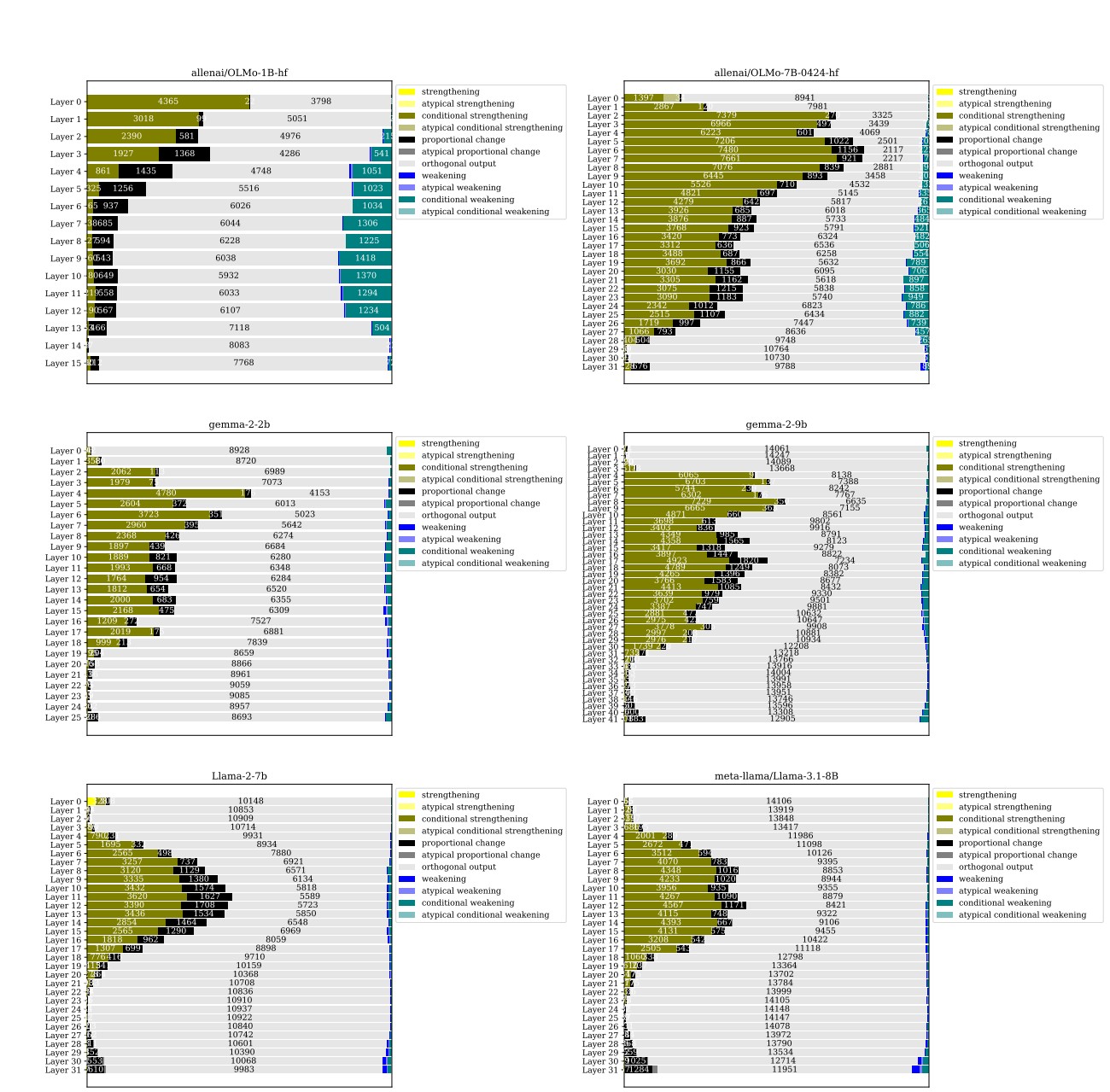

Figure 41: Distribution of neurons by layer and category for a range of models

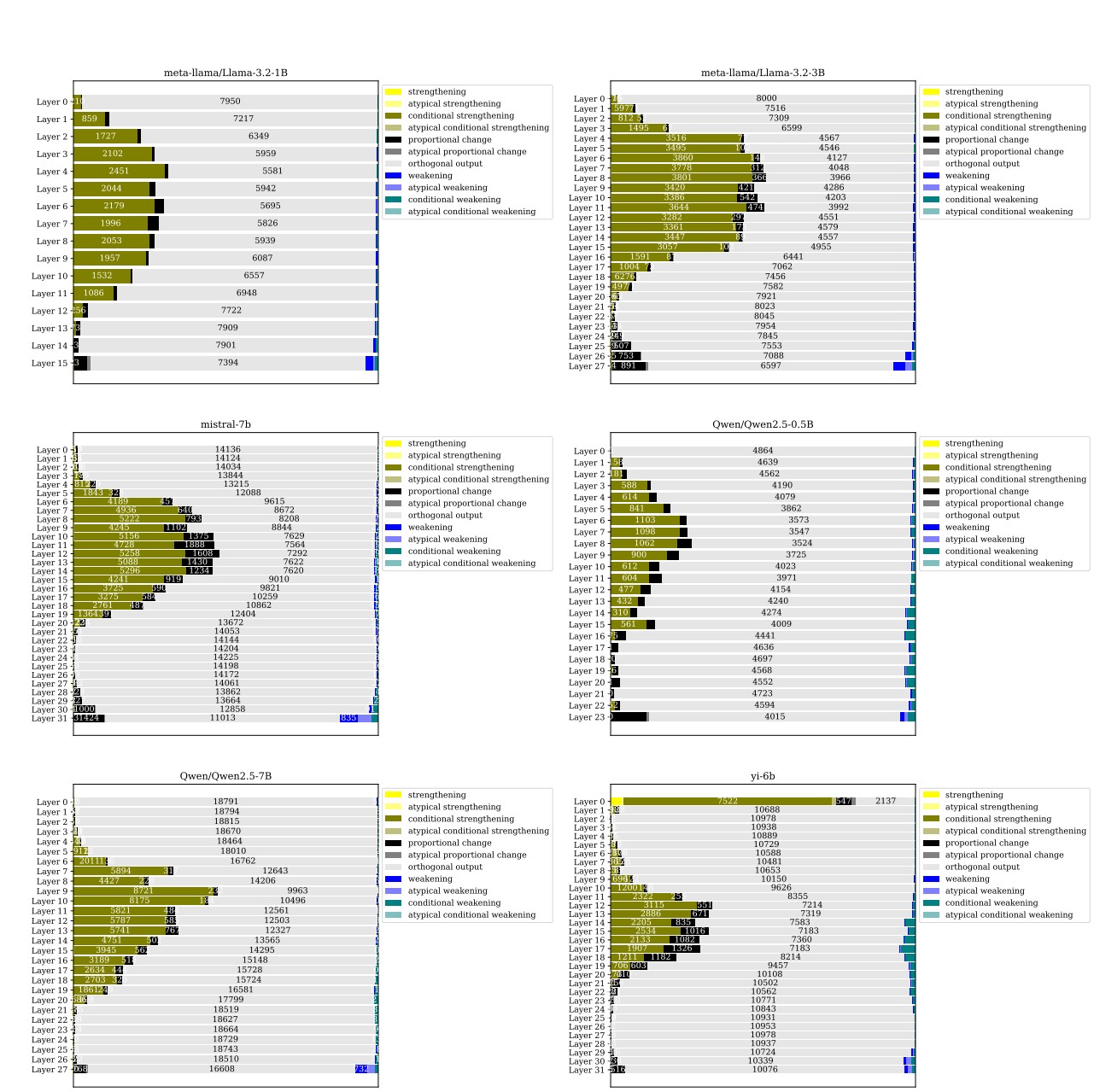

Figure 42: Continuation of figure 41. Including a copy of figure 1(b) (Llama-3.2-3B) for convenience.

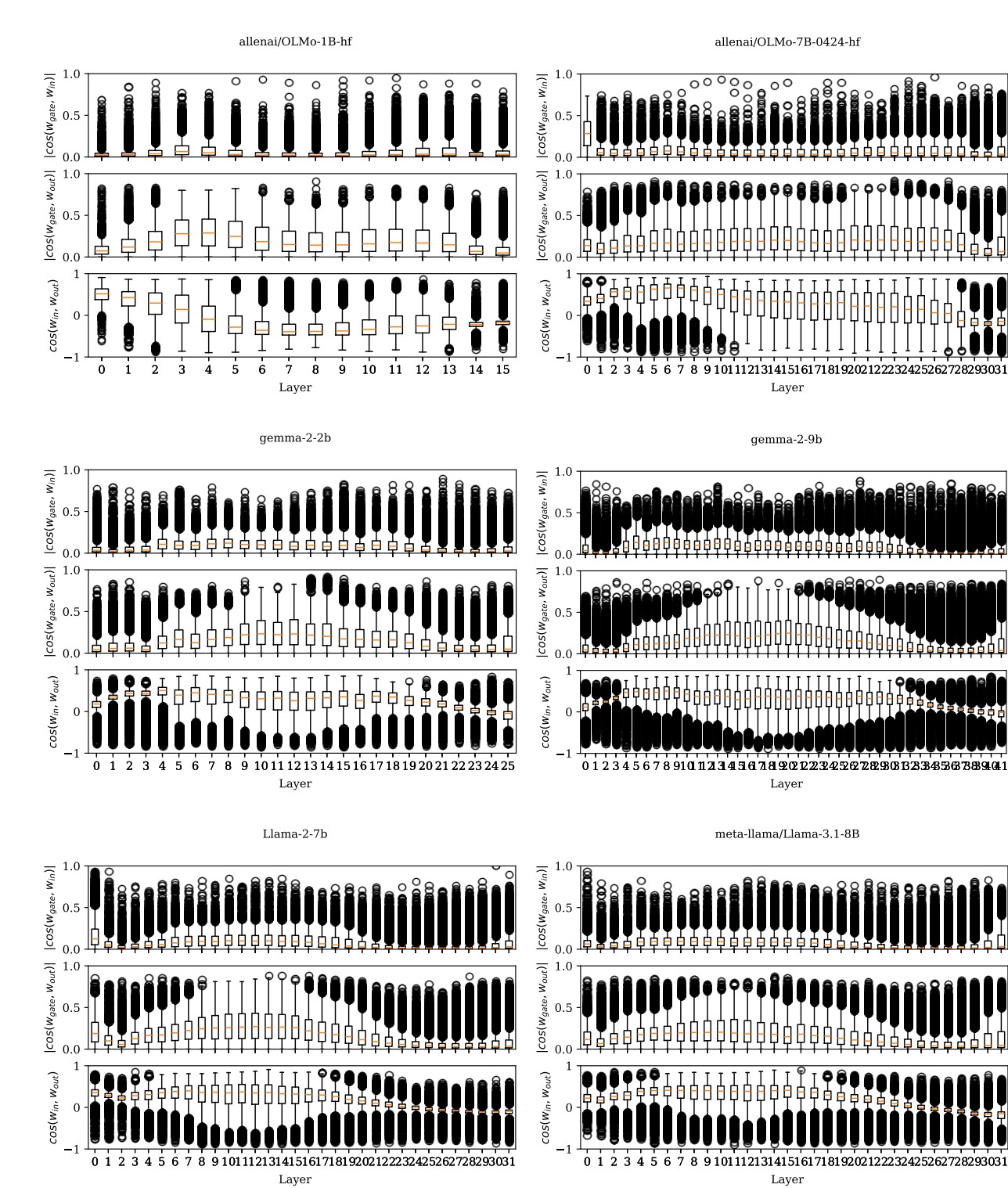

Figure 43: Boxplots for the distribution of weight cosine similarities in each layer. For $\cos(\boldsymbol{w}_{\text{gate}}, \boldsymbol{w}_{\text{in}})$ and $\cos(\boldsymbol{w}_{\text{gate}}, \boldsymbol{w}_{\text{out}})$ we show the absolute value since their sign does not carry any information on its own.

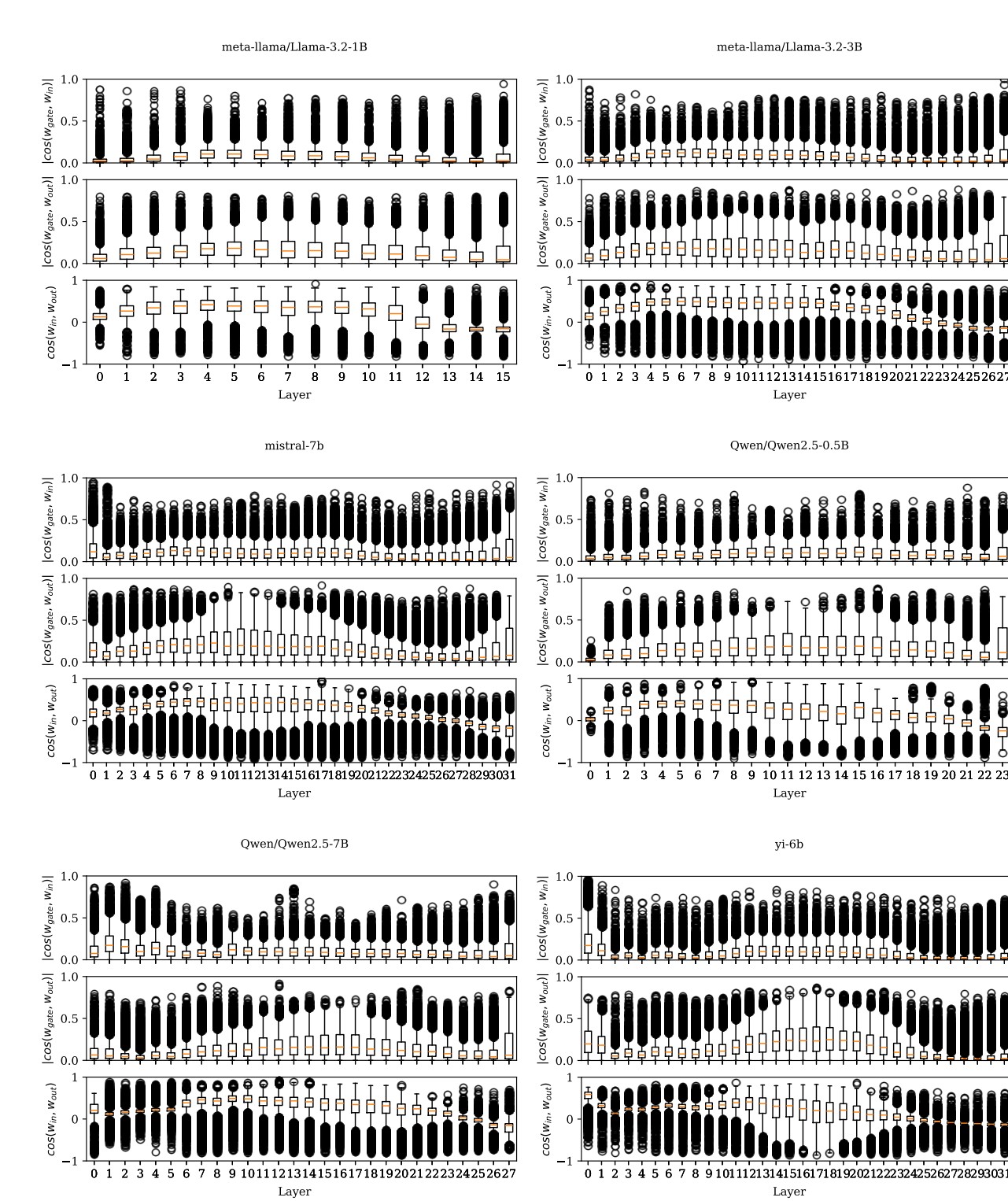

Figure 44: Continuation of figure 43.

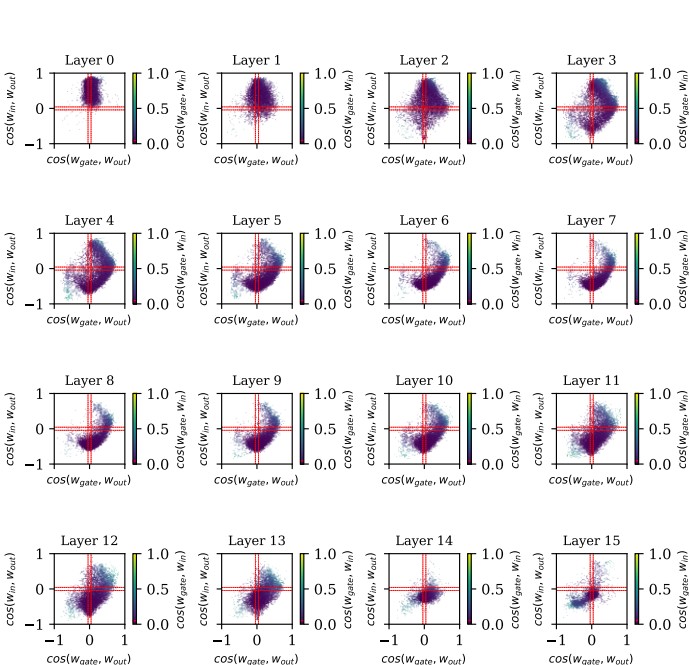

Figure 45: Equivalent of figure 2 for OLMo-1B

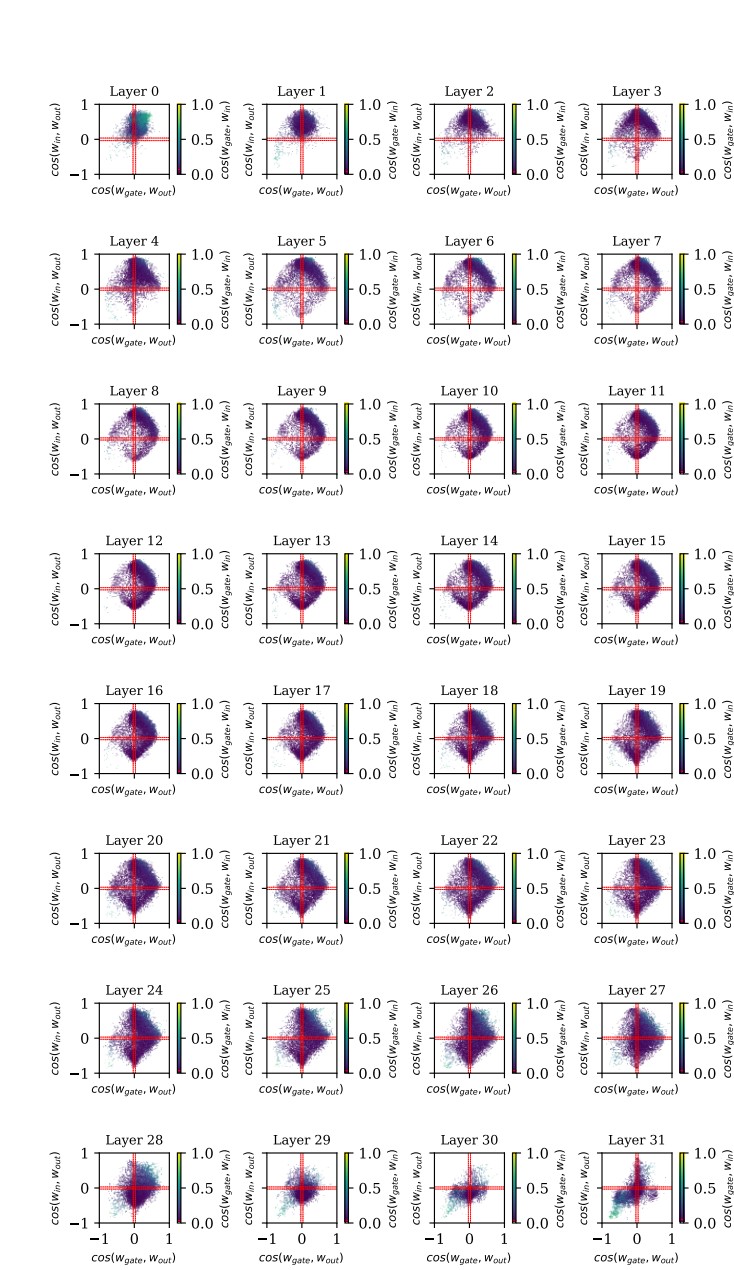

Figure 46: Equivalent of figure 2 for OLMo-7B-0424

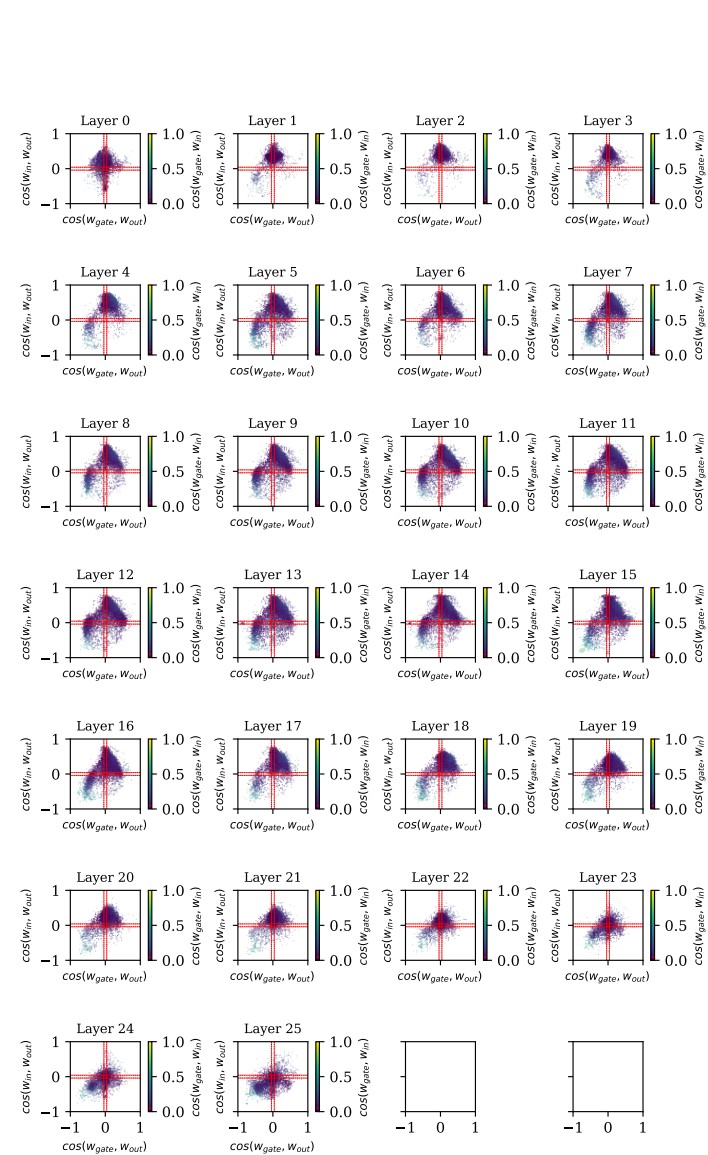

Figure 47: Equivalent of figure 2 for Gemma-2-2B

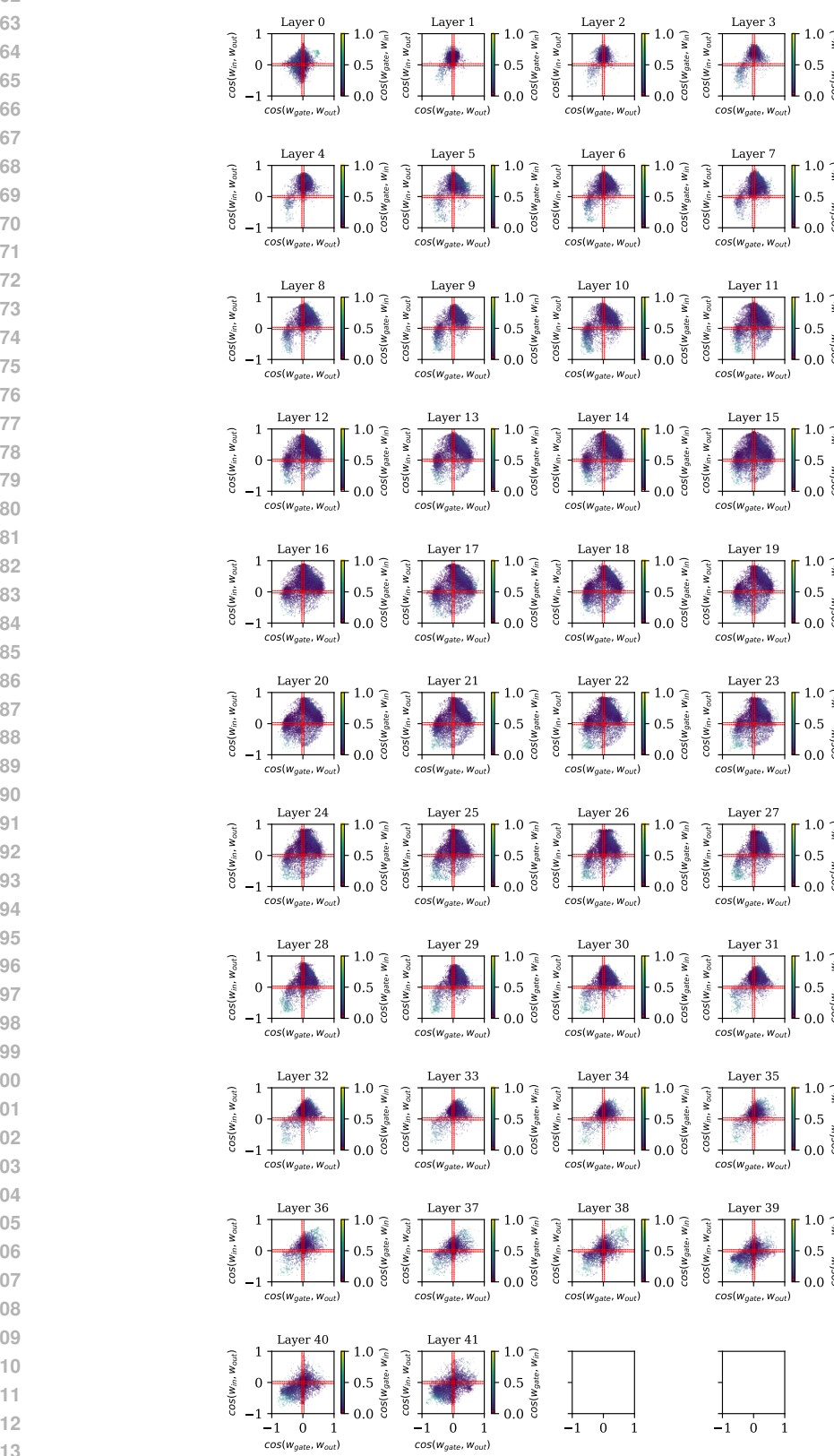

Figure 48: Equivalent of figure 2 for Gemma-2-9B

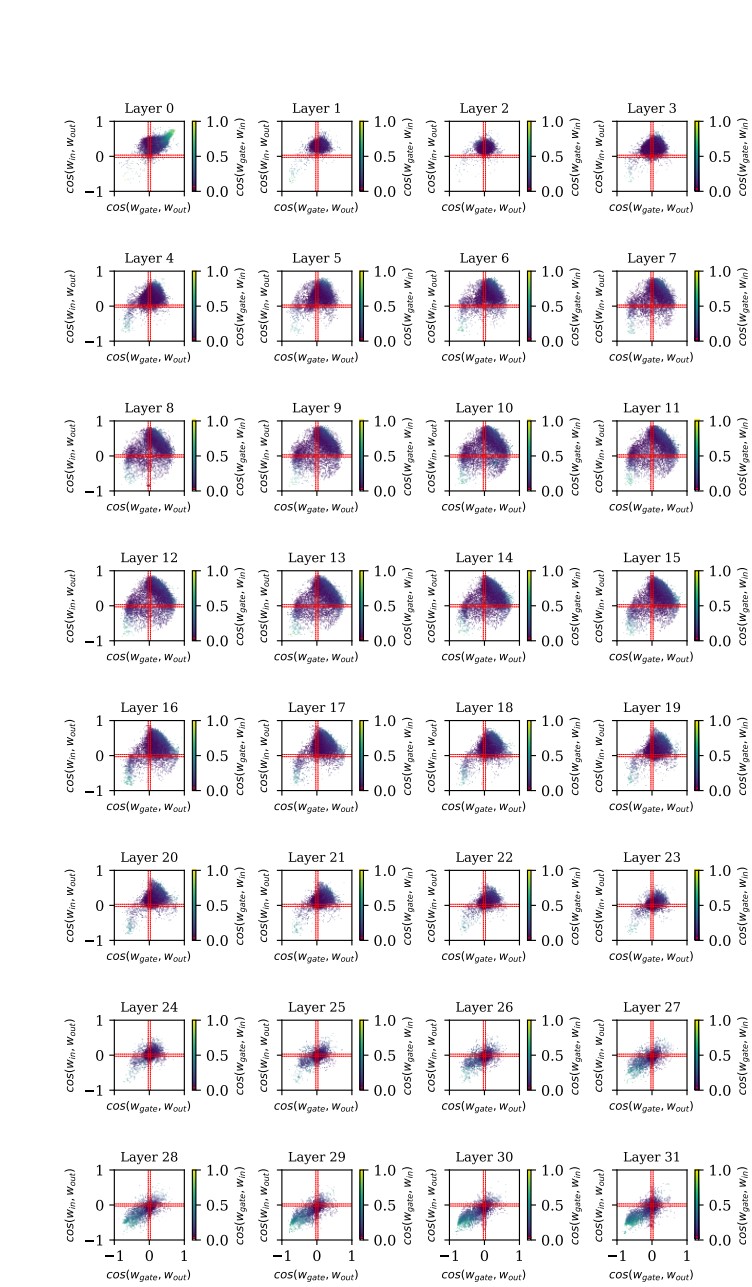

Figure 49: Equivalent of figure 2 for Llama-2-7B

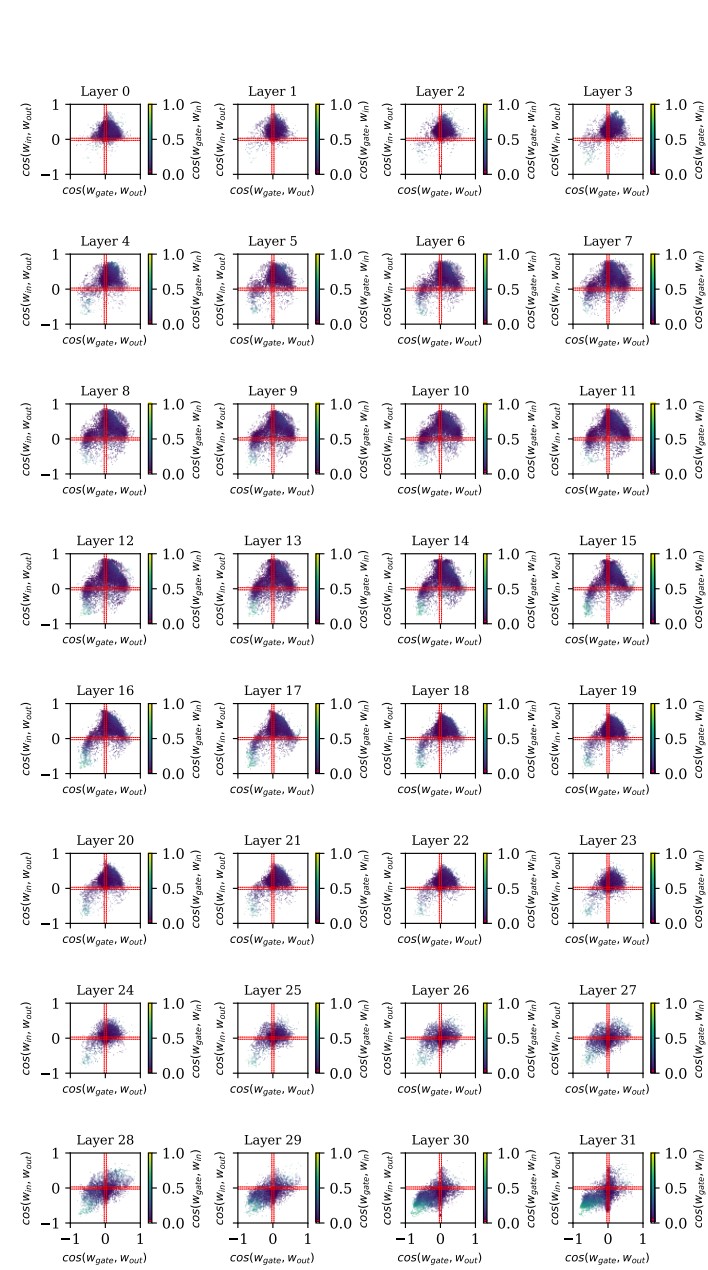

Figure 50: Equivalent of figure 2 for Llama-3.1-8B

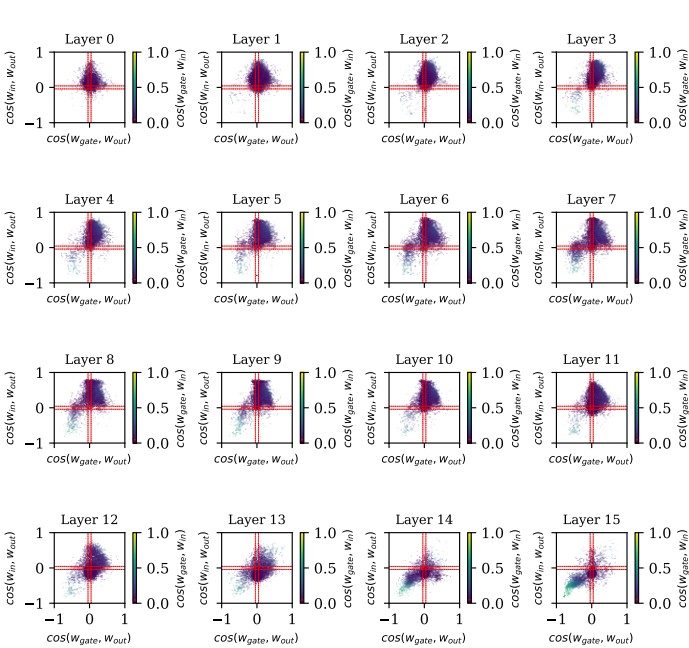

Figure 51: Equivalent of figure 2 for Llama-3.2-1B

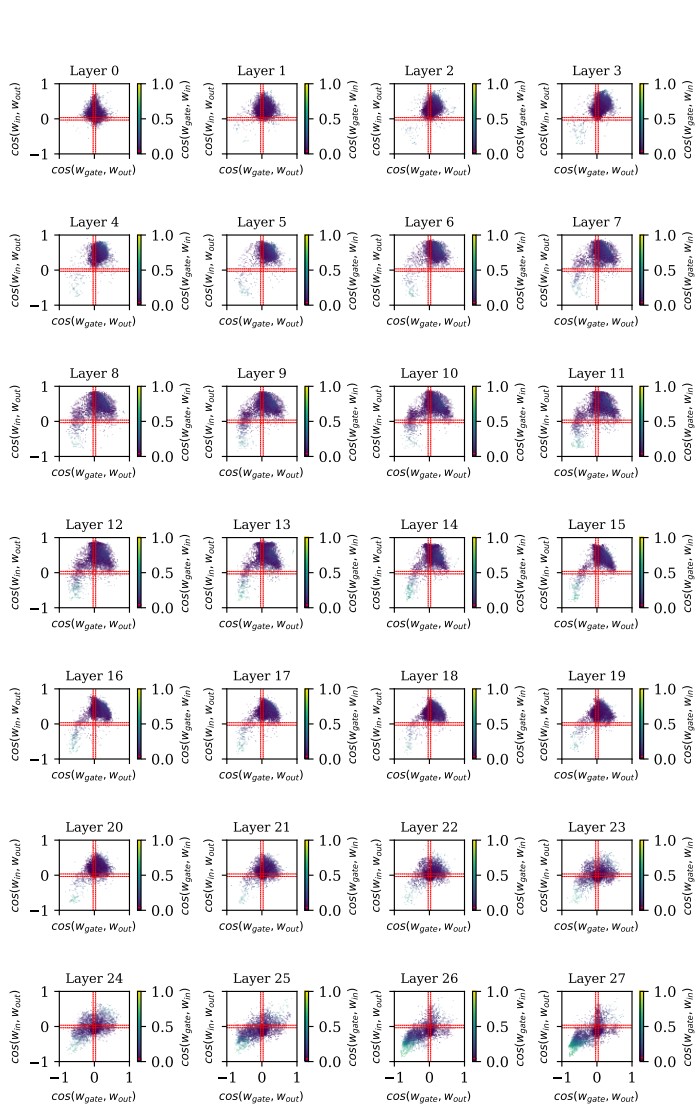

Figure 52: Equivalent of figure 2 for Llama-3.2-3B (same model but all layers)

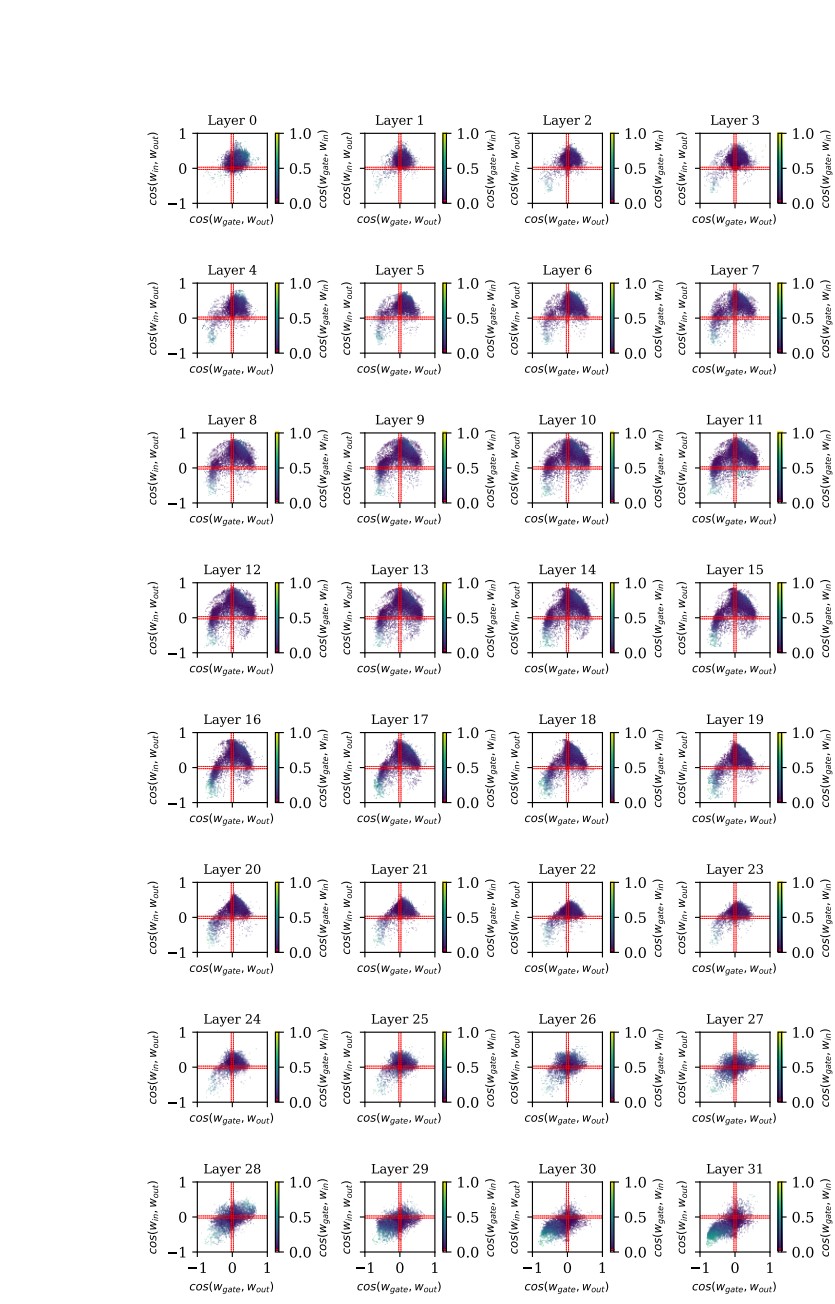

Figure 53: Equivalent of figure 2 for Mistral-7B

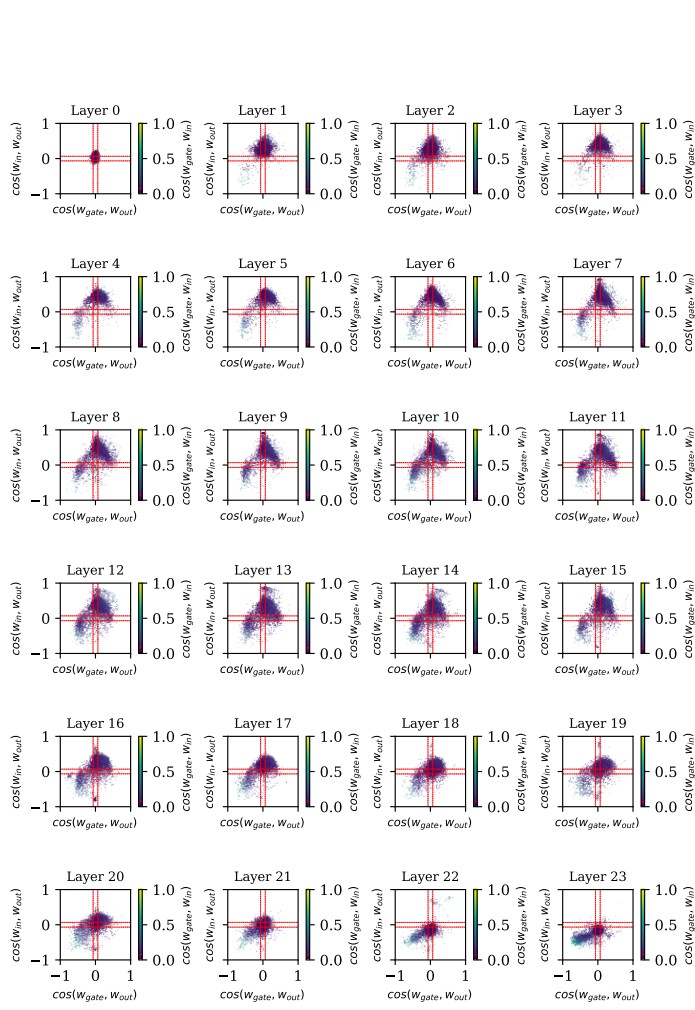

Figure 54: Equivalent of figure 2 for Qwen2.5-0.5B

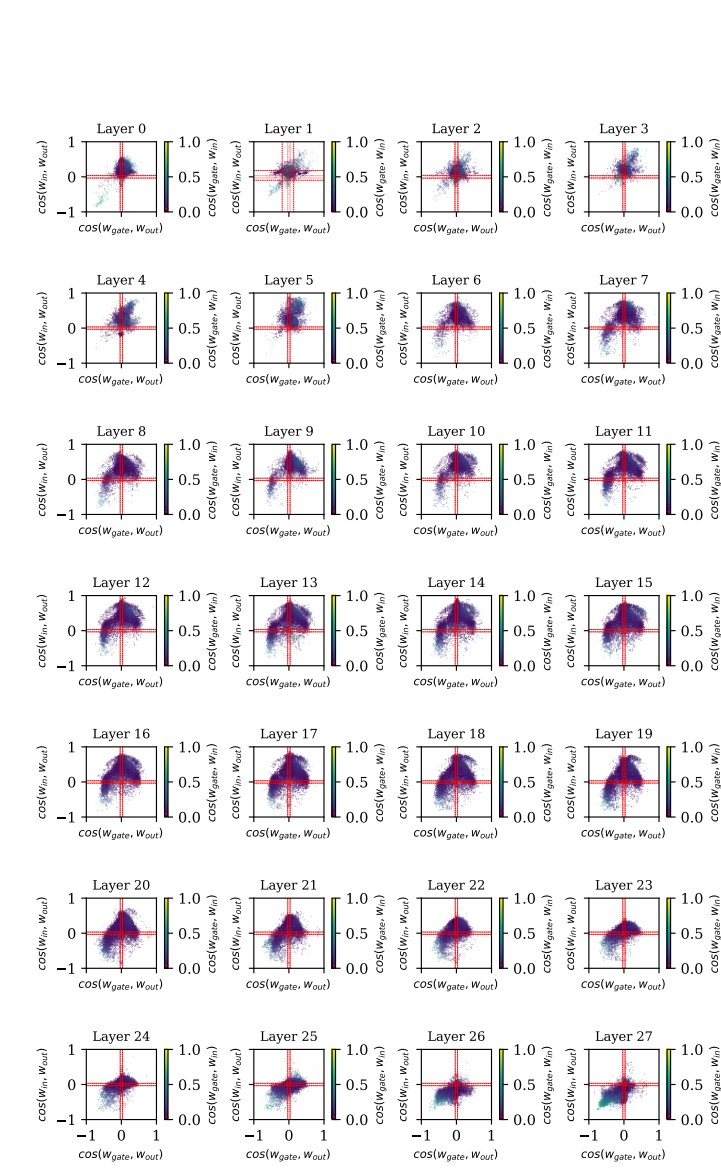

Figure 55: Equivalent of figure 2 for Qwen2.5-7B

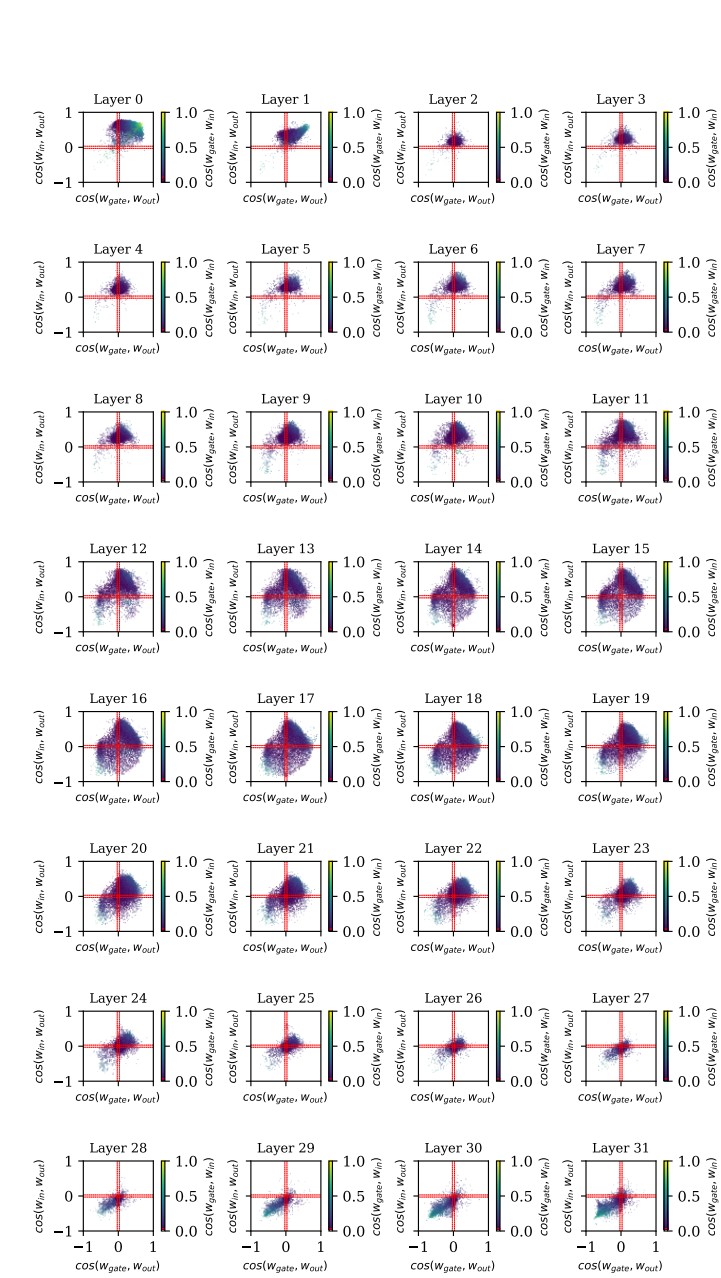

Figure 56: Equivalent of figure 2 for Yi-6B

