# OpenReview forum: "Weakening Neurons: A Newly Discovered Read-Write Functionality in Transformers with Outsize Influence"
_ICLR.cc/2026/Conference — Submitted to ICLR 2026_

### Official Review · Reviewer_kvb5 · 2025-10-20

**Soundness:** 3
**Presentation:** 1
**Contribution:** 2
**Rating:** 4
**Confidence:** 4

**Summary:**

This paper investigates a new phenomenon, weakening neurons, which occur in MLPs that use the SwiGLU activation function (as most LLMs currently do). These neurons appear in later layers, activate often, and write in the opposite direction as they read; this holds across a variety of models. Ablation experiments indicate that these neurons play a role in sharpening the model's output distribution. In addition to introducing these weakening neurons, this paper discusses other sorts of neurons, such as strengthening neurons, which are involved in early-mid layers.

**Strengths:**

This paper digs into the ways in which specific neurons act to shape the model's output distribution. It's long been hypothesized that later MLPs work to refine the model's output distribution, so it's interesting to understand how exactly models do this. The methods used are quite simple, but this isn't a bad thing - it keeps the paper clear, and simple causal methods are often good enough. The overall significance of this is unclear, but I think this work will be interesting to some mechanistic interpretability researchers.

**Weaknesses:**

Though I think this paper makes an interesting and valuable contribution, I also think that it does not yet meet the bar of an ICLR paper.

The primary issue is that the paper is sparse: it just doesn't have much content. It's 8.5 pages long, but if the figures were tightened up, it would easily be under 8 pages. These pages could be used to address real deficiencies in the paper. For example they could provide a case study in what exactly these weakening neurons do, as right now, discussion of these weakening neurons is entirely abstract. Instead of just noting that the distribution gets sharper, they could look at a specific example and show how these weakening neurons sharpen the distribution. I see that there is a section doing this in the appendix that goes entirely unmentioned in the main text - perhaps it's worth bringing that into the main text? Another good use of this space would be to study a model besides OLMo.

It's also worth engaging more with prior literature. In recent years, there has been quite a bit of debate about whether neurons are the right unit of analysis to study, and many possible solutions to related issues like polysemanticity / superposition. I don't think that these issues are totally critical for this paper, but it's worth engaging with that literature and thinking about how it might inform your results.

There are some minor issues too: unexplained choices (why use attributes score in Figure 4, and how is it measured?), hard-to-read figures (Figure 5 could be made horizontal, and more space could be given to the y-axis), and other such polish-related issues.

Finally, as a core mech interp researcher, I find this paper interesting. But, I think it could do a better job of motivating the question and methods used to study it - not just in terms of e.g. Gurnee et al.'s interest in this sort of mechanisms, but in terms of the broader impact of this question, at least for the interp community.

**Questions:**

- Why do you use attributes score in Figure 4, and how is it measured?
- How do you think your findings are affected by polysemanticity / superposition?

---

> ### Author Response · Authors · 2025-11-21
>
> Thank you for your review and your encouragement!
> We particularly appreciate that you took time to look at the appendix as well.
>
> In the updated version of the paper:
> * we try to improve clarity, addressing a general concern of all the reviewers;
> * we describe a neuron case study that was previously only in the appendix (new section 8), as you suggested;
> * we repeat the ablation experiments with mean ablation (new lines 352-357 and appendix F.4);
> * we present a text example in which negative gate values of weakening neurons lead to an especially large reduction in entropy (new section 6.3), addressing another one of your suggestions.
>
> For lack of time, we did not repeat the ablation experiments with another model.
>
> *How do you think your findings are affected by polysemanticity / superposition?*
>
> We now briefly justify the neuron-based approach in l. 173-180
> and the other passages referenced therein.
> In short:
> (i)
> we think there are still things to find out about neurons
> (and we think our paper is a good example of this!);
> (ii)
> we give a rough argument that a linear combination of, say, strengthening neurons
> will also have a behavior similar to strengthening.
>
> For additional thoughts on superposition:
> Our new text case study in section 6.3 shows some evidence
> that weakening neurons work together in superposition
> to achieve the observed effect.
> In the neuron case study of section 8
> (detailed in appendix I.3),
> we hypothesize that some of the activations of this weakening neuron
> help get rid of the unwanted side-effects of superposition.
>
> *Why do you use attribute scores in Figure 4, and how is it measured?*
>
> We talk about this in appendix F, especially F.1 and F.3.

---

> > ### Comment · Reviewer_kvb5 · 2025-11-24
> >
> > Thanks for all this! I appreciate you moving the case study into Section 8 (although more could be said there - remember that you have 10 pages during the rebuttal period!); I do think it improves the paper. I still feel that this paper is a bit sparse, and its wider importance is not super clear. I will keep my current score, while keeping in mind that a 4 means I *would not mind if paper is accepted*.

---

### Official Review · Reviewer_SWdj · 2025-10-31

**Soundness:** 2
**Presentation:** 2
**Contribution:** 3
**Rating:** 2
**Confidence:** 3

**Summary:**

This paper is an interpretability study of neurons in gated MLPs as used in modern LLMs, mainly by considering the angles between input, gate, and output vectors. Neurons are classified on the basis of the rough size/magnitude of the cosines between these angles. Among other findings, the key finding from the paper is that "weakening neurons" have substantial influence despite being not so large in number.

**Strengths:**

- Given how prevalent gated MLPs are in modern LLMs, and how the function of MLPs tends to still be much less clear than that of attention, a study elucidating their inner workings can be a welcome addition to the literature.
- The paper studies a substantial variety of models from different families and sizes, establishing some key conclusions (Section 3) across them.

**Weaknesses:**

I believe the current version has the following weaknesses. I'll be happy to reconsider my score based on the response.

Rigor-related weaknesses:
- Section 4.1: Ablations are done using zero ablation. Zero ablations are quite destructive and may move the model activations out-of-distribution. Could the outsized effect of weakening neurons somehow be caused by a confound related to this ablation method? Could the same effect be shown with a less destructive ablation method (e.g., mean ablation)?

Clarity-related weaknesses:
- line 114: "Additionally, it guarantees that two equivalent neurons by property (2) will now also have the same weights" -- I don't understand what this means
- line 137-142: I found this paragraph confusing and didn't understand what it aims to express. What is "atypical" here meant to express?
- Section 3: What is the input data? I didn't see this specified. I'm also confused that line 262 states that the model is run on a dataset in Section 4, which suggests Section 3 doesn't involve running the model on any data?
- Figure 4: Why is there no "clean" curve even though it appears in the caption?

**Questions:**

- I have one question about how exactly the alignment between vector directions is defined.
-- line 124 has $\approx \pm 1$, whereas Table 1 has $>> 0$, $<<0$. Can the authors comment on whether there is an inconsistency here?
-- line 126: "roughly collinear" is this supposed to mean the same as $cos(...) \approx \pm 1$? It could improve readability of the paragraph to streamline terminology.

Minor suggestions:
- Since Swish isn't as broadly familiar as ReLU, but is central to the paper, it couldn't hurt introducing it in Section 1.2 explicitly
- Figure 2: this is very small
- page 6: there is a huge waste of space, which the authors could use more efficiently

---

> ### Author Response · Authors · 2025-11-21
>
> Thank you for your review!
>
> In the updated version of the paper:
> * we try to improve clarity, addressing a general concern of all the reviewers;
> * we describe a neuron case study that was previously only in the appendix (new section 8);
> * we repeat the ablation experiments with mean ablation (new lines 352-357 and appendix F.4), addressing one of your concerns;
> * we present a text example in which negative gate values of weakening neurons lead to an especially large reduction in entropy (new section 6.3).
>
> *line 114: "Additionally, it guarantees that two equivalent neurons by property (2) will now also have the same weights" -- I don't understand what this means*
>
> We now just briefly mention our weight processing in section 3.2.
> It is just a technicality and not part of our core method,
> so we moved a detailed discussion of it to appendix C.
> We hope these detailed explanations make more sense to you.
>
> *line 137-142: I found this paragraph confusing and didn't understand what it aims to express. What is "atypical" here meant to express?*
>
> This is now lines 208-254.
> We hope the paragraph is clearer now.
> In table 1 and the corresponding lines 196-211,
> we just looked at $\cos(w_\text{in}),w_\text{out})$ and $\cos(w_\text{gate},w_\text{out})$,
> but there is a third cosine similarity to consider: $\cos(w_\text{gate},w_\text{in})$.
> This third cosine can have weird patterns with respect to the other two:
> for example when $w_\text{in}$ and $w_\text{gate}$ are both highly similar to $w_\text{out}$,
> but not to each other.
> These are the cases we call "atypical".
>
> *Section 3 doesn't involve running the model on any data?*
>
> This is correct:
> This section (section 4 in the new version)
> is a purely weight-based analysis,
> where we just apply the cosine similarity method of the previous section to all the LLM neurons.
> The fact that the results were found with such a simple method
> makes them all the more striking in our opinion.
> In the new version we tried to emphasize this more,
> at the beginning of this section and in the paper more generally.
>
> *Figure 4: Why is there no "clean" curve even though it appears in the caption?*
>
> The clean curve was hidden under the baseline curve.
> We have now improved the plot layout so that this becomes visible (new fig. 3a).
>
> *"$\approx \pm 1$" vs "$\ll 0$" vs "roughly collinear"*
>
> We agree this was confusing.
> In our theoretical reflection of l. 187-202 (new version)
> we first reflect on what we call prototypical cases,
> with cosine similarities roughly $\pm 1$ (equivalent to "roughly collinear")
> or roughly 0 (orthogonal).
> Actual weights are of course somewhere in between,
> and l. 208-254 discuss possible ways to deal with this.
> One of these ways is threshold-based classification (with a threshold of $\pm 0.5$).
> We have now included this thresholding option in table 1 as well,
> and got rid of the confusing "$\ll 0$" expression.
> We hope this makes things clearer
>
> *introduce Swish explicitly*
>
> Done in l. 108-112 of the new version.

---

> > ### Comment · Reviewer_SWdj · 2025-11-21
> >
> > Thanks for the clear rebuttal, which thoroughly addresses my concerns. I'm updating my scores.

---

### Official Review · Reviewer_ZwoJ · 2025-11-01

**Soundness:** 4
**Presentation:** 3
**Contribution:** 4
**Rating:** 8
**Confidence:** 3

**Summary:**

The authors identify neurons that conditionally write (strengthen, weaken) concepts. They study the behavior of these neurons in depth.

**Strengths:**

The paper includes baselines and conducts a large number of experiments to identify and understand a novel type of behavior in LLMs. It is especially surprising how universal this mechanism seems to be. I appreciated the section on the intuition the authors gained.

**Weaknesses:**

The presentation could benefit from being more crisp about what the claims are. In addition, it was unclear to me why finding negative gating was meaningful.

**Questions:**

Could you include concrete examples in your presentation? Of certain neurons and their function based on your analysis. Right now it felt very abstract, it would be beneficial to have particular case studies that highlight the general trend.

---

> ### Author Response · Authors · 2025-11-21
>
> Thank you for your review!
>
> In the updated version of the paper:
> * we try to improve clarity, addressing a general concern of all the reviewers;
> * we describe a neuron case study that was previously only in the appendix (new section 8), addressing one of your concerns;
> * we repeat the ablation experiments with mean ablation (new lines 352-357 and appendix F.4);
> * we present a text example in which negative gate values of weakening neurons lead to an especially large reduction in entropy (new section 6.3).
>
> *why finding negative gating was meaningful*
>
> See the new lines 377-381.
> Activation functions like Swish (or GELU) are often seen as smooth approximations of ReLU,
> and with ReLU negative values don't matter.
> The fact that such functions work better than ReLU was often attributed to just training dynamics.
> We find that negative values do matter,
> not just in training but also in the mechanisms of the final model.
> This is relevant for future neuron analysis research:
> it is not enough to treat these functions like ReLU as was often done previously.

---

### Meta-Review · Area_Chair_hgZs · 2025-12-27

**Summary:**

Reviewers found the paper interesting, but raised several concerns that, taken together, led to the AC’s recommendation to reject. First, while the paper reports that a small subset of “weakening neurons” has a disproportionately large effect on model behavior, the evidence appears sensitive to the intervention choice. In particular, the effects observed under zero ablation become less pronounced and less clearly separable when mean ablation, as requested by **Reviewer Swdj**, is considered, making the strength of the resulting conclusions more difficult to assess. **Reviewer kvb5** further noted that the paper is relatively sparse given its length: key claims are presented at a high level of abstraction, with limited concrete case studies or mechanistic depth in the main text. Finally, while the application of weight-space cosine analysis to gated MLPs is interesting, the paper does not sufficiently relate or distinguish its contributions from prior work on neuron roles, MLP functionality, and suppression or sharpening mechanisms in transformers.

**Reviewer Concerns:**

The rebuttal and revised manuscript addressed several **clarity** concerns raised by the reviewers. It also introduced additional ablation experiments using mean ablation in response to Reviewer **SWdj**’s concern about zero ablation. Reviewer **SWdj** explicitly mentioned that the rebuttal addressed their specific questions. That said and despite the inclusion of a case study, reviewer **kvb5** did not champion the paper and commented that core claims of the paper remain largely at a high level and limited mechanistic depth in the main text.

**Reviewer Scores:**

Based on the rebuttal and subsequent discussion, the AC expects that Reviewer **SWdj** would increase their score modestly, likely from 2 to around 4, but not to a level that would strongly support acceptance. Reviewer **ZwoJ**, who was broadly positive, would likely maintain their high score; however, it is also plausible that discussion focusing on robustness and novelty could lead to a slightly more cautious assessment (moving from an 8 to a weaker accept around 6). Reviewer **kvb5**, who expressed the highest confidence in their evaluation, would likely maintain their score of 4.

---

### Decision · Program_Chairs · 2026-01-26

Reject